# Structural insight into the activation of a class B G-protein-coupled receptor by peptide hormones in live human cells

Lisa Seidel[1], Barbara Zarzycka[2], Saheem A Zaidi[2], Vsevolod Katritch[2,3]*, Irene Coin[1]*

[1]Institute of Biochemistry, Leipzig University, Leipzig, Germany; [2]Department of Biological Sciences, Bridge Institute, University of Southern California, Los Angeles, United States; [3]Department of Chemistry, Bridge Institute, University of Southern California, Los Angeles, United States

**Abstract** The activation mechanism of class B G-protein-coupled receptors (GPCRs) remains largely unknown. To characterize conformational changes induced by peptide hormones, we investigated interactions of the class B corticotropin-releasing factor receptor type 1 (CRF1R) with two peptide agonists and three peptide antagonists obtained by N-truncation of the agonists. Surface mapping with genetically encoded photo-crosslinkers and pair-wise crosslinking revealed distinct footprints of agonists and antagonists on the transmembrane domain (TMD) of CRF1R and identified numerous ligand-receptor contact sites, directly from the intact receptor in live human cells. The data enabled generating atomistic models of CRF- and CRF(12-41)-bound CRF1R, further explored by molecular dynamics simulations. We show that bound agonist and antagonist adopt different folds and stabilize distinct TMD conformations, which involves bending of helices VI and VII around flexible glycine hinges. Conservation of these glycine hinges among all class B GPCRs suggests their general role in activation of these receptors.

**\*For correspondence:** katritch@usc.edu (VK); irene.coin@uni-leipzig.de (IC)

**Competing interests:** The authors declare that no competing interests exist.

## Introduction

G-protein-coupled receptors (GPCRs) of class B comprise a family of 15 transmembrane receptors that respond to endocrine factors and regulate vital functions in mammals, including glucose and calcium homeostasis, pain transmission and gastrointestinal regulation (*Bortolato et al., 2014*; *Wootten et al., 2017*; *Culhane et al., 2015*; *Pal et al., 2012*). While a number of clinical drugs already employ class B GPCRs to treat hypercalcemia, osteoporosis, diabetes and eating disorders, many new therapies targeting these receptors are being pursued (*Bortolato et al., 2014*; *Wootten et al., 2017*). In addition to the heptahelical architecture of the transmembrane domain (TMD) common to all GPCRs, class B receptors are defined by a large, glycosylated N-terminal extracellular domain (ECD, ~120–160 residues in length) (*Culhane et al., 2015*; *Pal et al., 2012*). The native ligands of class B GPCRs are long peptide hormones, which bear distinct functional sites at the two termini (*Pal et al., 2012*; *Beyermann et al., 2000*). The C-terminal portion of the peptide forms selective high-affinity interactions with the ECD and drives selective receptor binding, while the N-terminal segment interacts with the TMD and triggers receptor activation (two-domain binding model) (*Culhane et al., 2015*; *Pal et al., 2012*; *Dong et al., 2014*; *Beyermann et al., 2000*).

While recent structural studies have provided major insights into activation of class A GPCRs by small molecule ligands, little is known about activation mechanisms of class B GPCRs by peptide hormones, despite availability of some structural information for these receptors. Thus, crystal structures have revealed the binding mode of ligand C-termini to the isolated ECDs of several class B

receptors, but they excluded TMDs (*Bortolato et al., 2014*; *Pal et al., 2012*). TMD structures for the glucagon receptor and the corticotropin-releasing factor (CRF) receptor type 1 (CRF1R) were also solved, but they lack the ECDs and represent the receptors in an inactive state bound to allosteric small molecule inhibitors (*Hollenstein et al., 2013*; *Jazayeri et al., 2016*; *Siu et al., 2013*). Most recently, the full-length crystal structure of the glucagon receptor in complex with a small molecule acting as negative allosteric modulator (*Zhang et al., 2017*) and the cryo-electron microscopy (cryo-EM) structure of the calcitonin receptor in complex with a Gs-protein heterotrimer were also solved (*Liang et al., 2017*). While the complex used for cryo-EM included a peptide agonist and the ECD, high flexibility prevented structure determination in this part of the complex, thus highlighting the limitations of structural approaches that require conformational rigidity in the protein, and leaving interaction modes of peptide agonists and antagonists with full-length class B GPCRs unresolved.

It is a general feature of class B natural agonists that they turn into partial agonists and then into antagonists by successively truncating residues at the N-terminus. However, the molecular basis of this phenomenon has never been investigated. We reasoned that comparing the binding mode of class B agonists and orthosteric N-truncated antagonists would shed light on conformational changes in the receptor TMD that lead to activation and may facilitate drug discovery at these receptors.

As prototype class B receptor, we used the CRF1R, which is a key regulator in stress response by triggering the release of adrenocorticotropic hormone (ACTH) in the pituitary gland (*Bale and Vale, 2004*; *Stengel and Taché, 2014*). Small molecule and peptide-based clinically applicable antagonists of CRF1R have long been sought after for treating stress-related disorders such as anxiety and depression (*Liapakis et al., 2011*; *Grigoriadis and Hoare, 2017*). In humans, CRF1R is activated by the two 41- and 40-mer endogenous peptide agonists CRF and Urocortin I (Ucn1). N-terminal truncation of the first eight residues of CRF yields antagonists retaining some weak partial activity, such as CRF(9-41) (*Rivier et al., 1984*; *Rivier and Rivier, 2014*). Signaling is completely lost by truncating 11 residues, as in [DPhe$^{12}$, Nle$^{21,38}$]-CRF(12-41) and Astressin (*Gulyas et al., 1995*). Although we have recently revealed the binding mode of the agonist Ucn1 on the CRF1R (*Coin et al., 2013*, *2011*), receptor interactions with peptide antagonists remain enigmatic.

In this study, we characterized the CRF1R complexes with the agonist CRF and three N-truncated peptide antagonists. We used unnatural amino acid photo-crosslinking and pair-wise chemical crosslinking to reveal ligand-receptor interactions in live cells. We defined ligand footprints on the CRF1R and derived sets of intermolecular contacts, which were applied in extensive conformational sampling to generate molecular models of agonist- and antagonist-bound CRF1R. We found that CRF1R peptide antagonists stabilize different conformations of the CRF1R TMD in respect to the agonists, including large-scale movements of transmembrane helices and changes in the shape of the binding pocket.

## Results

### Footprints of agonists and antagonists on CRF1R

In first place, we mapped agonist and antagonist footprints on CRF1R in intact 293T cells using the photo-crosslinking amino acid p-azido-Phe (Azi) as a proximity probe.

We compared the binding paths of five peptide ligands: the natural agonists CRF and Ucn1, the two 33-mer antagonists CRF(9-41) and Ucn1(8-40), and the 30-mer antagonist [DPhe$^{12}$,Nle$^{21,38}$]-CRF (12-41) (*Figure 1A*), which from now on we will refer to as $^{dFX}$CRF(12-41). Activity of all peptides was assessed via cAMP accumulation assays and showed good agreement with literature data (*Table 1* and *Figure 1B*). All truncated peptides right shifted the dose-response curve of their parent agonist and behaved as competitive antagonists (*Figure 1C–D*).

Azi was incorporated into CRF1R in response to an amber stop codon using a dedicated aminoacyl tRNA synthetase/tRNA$_{CUA}$ pair (*Coin et al., 2013*). The receptor was equipped with a FLAG-tag at the C-terminus. We systematically mapped 119 positions spanning the whole juxtamembrane region of CRF1R, with the sole exclusion of eight known non-tolerant sites (*Coin et al., 2013*). Cell-surface expression of a representative subset of Azi-CRF1R mutants was determined using both immunoblotting of the C-terminal FLAG and a whole-cell ELISA detecting an ad hoc introduced N-terminal HA epitope (*Figure 1—figure supplement 1*). We observed a good correlation between

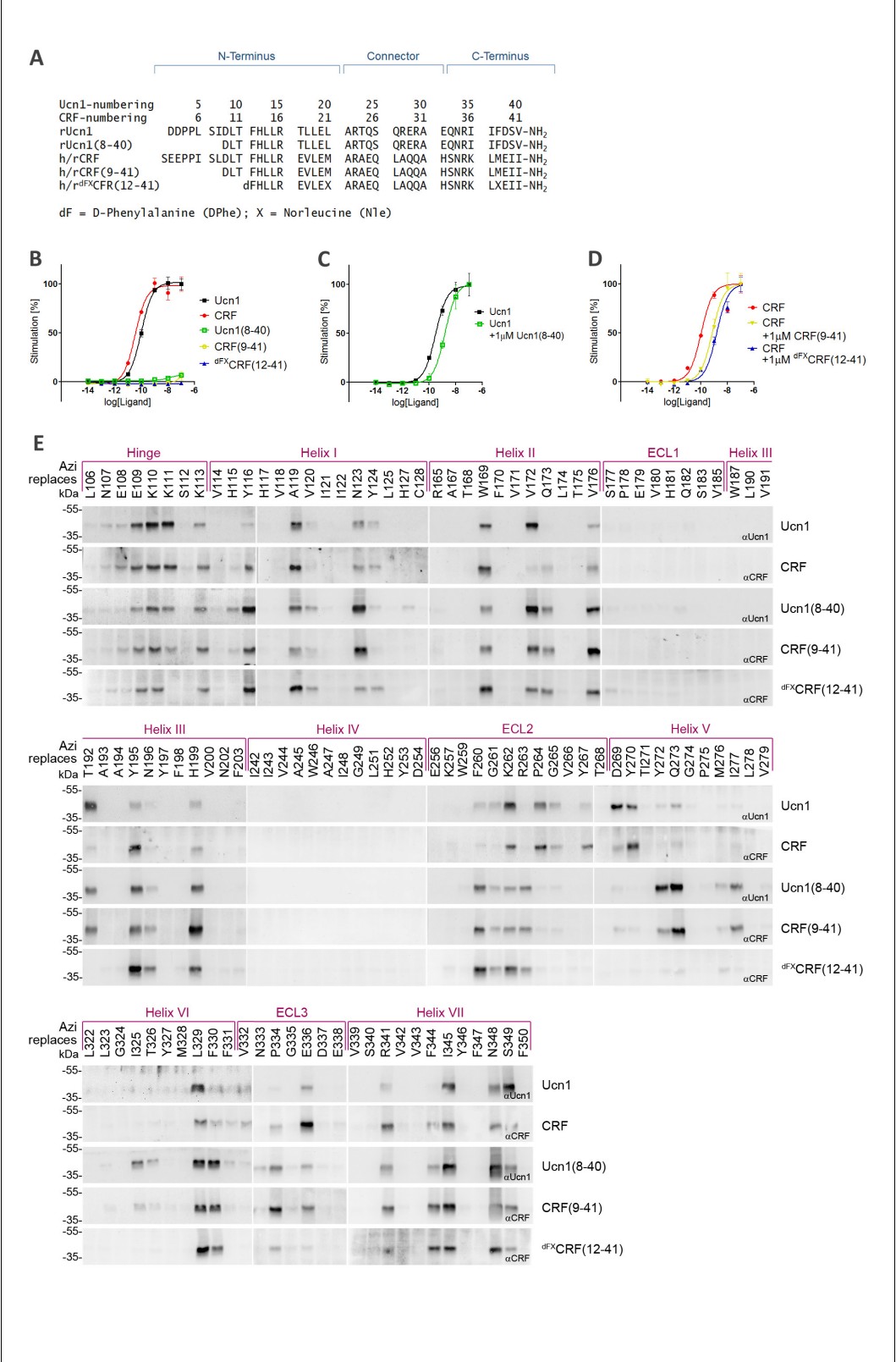

**Figure 1.** Photo-crosslinking mapping of CRF1R to determine footprints of agonists and antagonists. (**A**) Sequence alignment of CRF1R peptide ligands. The classification into N-terminus, C-terminus and helical connector is adopted from *Beyermann et al. (2000)*. Antagonists derived from CRF are well characterized in the literature (*Rivier and Rivier, 2014*), while Ucn1(8-40) was introduced in this work for the first time for direct comparison of agonist/antagonist pairs. In $^{dFX}$CRF(12-41), substitutions of the two native Met residues to Nle and the N-terminal D-amino acid to enhance its stability

*Figure 1 continued on next page*

*Figure 1 continued*

and potency (**Rivier and Rivier, 2014**). (B–D) Activation of CRF1R by CRF and Ucn1 in the presence and absence of competitive antagonists. The assay measures cAMP accumulation in live HEK293 cells stimulated with serial dilutions of each ligand via a luciferase reporter. Plots are representative mean ± s.e.m. of at least three independent experiments, each run in triplicates. (B) CRF1R activation elicited by each of the ligands used in this study. Sub-nanomolar $EC_{50}$ values for the two agonists CRF and Ucn1 agree with literature data (**Rivier and Rivier, 2014**). As expected, [dFX]CRF(12-41) did not elicit any significant receptor activation up to the maximal tested concentration of 100 nM, while CRF(9-41) and Ucn1(8-40) showed minimal residual activity (<10%). (C, D) CRF1R activation by either CRF or Ucn1, in the presence of a constant concentration of each of the antagonists used in this study. (E) Photo-crosslinking experiments. Transiently transfected 293T cells expressing each Azi-CRF1R mutant (residues replaced by Azi are indicated in the upper row) were incubated with each of the five peptide ligands indicated in the right column, followed by UV irradiation (365 nm). Whole-cell lysates were separated on 10% SDS-PAGE and analyzed by Western blotting using either anti-CRF or anti-Ucn1 antibodies as indicated. To obtain sharper bands, samples were deglycosylated by PNGaseF. The subdivision of transmembrane helices and ECLs is based on the crystal structure of the CRF1R TMD (**Hollenstein et al., 2013**). The deglycosylated ligand-CRF1R complex runs at an apparent MW of ~40 kDa (**Coin et al., 2013**). The non-crosslinked ligand is not detected (MW ~3–4 kDa).

The following source data and figure supplement are available for figure 1:

**Source data 1.** Numerical values of cAMP accumulation assay.

**Source data 2.** Numerical values of whole-cell ELISA of HA-Azi-CRF1R mutants.

**Figure supplement 1.** Expression profiles of a subset of Azi-CRF1R mutants.

the amount of mature glycosylated receptors detected in Western blot and the ELISA signals obtained through detection of the extracellular tag. All mutants showed expression levels in the same order of magnitude, with variations of 0.3–0.9-fold with respect to the wild-type receptor.

Cells expressing each mutant receptor were treated with each of the five ligands and crosslinking was triggered with UV light. If the bound ligand lies within the radius of reach of Azi (~9 Å from $C\beta$, see also modeling section below), it can be captured by the photo-active moiety (**Coin et al., 2013**). A covalent ligand-receptor complex is formed, which is detected in Western blot at the approximate molecular weight of the receptor with an anti-ligand antibody. Occurrence of crosslinking was detected at numerous positions distributed along the whole TMD with all ligands (**Figure 1E**). Residue positions are indicated with Wootten numbering (**Wootten et al., 2013**) for class B GPCRs, as implemented in GPCRdb (**Isberg et al., 2015**).

Overall, we observed two different patterns of crosslinking footprints: one shared by the two agonists and a different one for the three antagonists (**Figure 2, Figure 2—figure supplement 1**). The hits obtained with the agonist CRF overlapped with those of the agonist Ucn1. Likewise, the crosslinking hits of the 33-mer antagonists CRF(9-41) and Ucn1(8-40) were almost identical to each other. The shorter antagonist [dFX]CRF(12-41) shared the pattern of the 33-mer antagonists with the exception of helix V, where it lacked any hits. Agonist and antagonist footprints differed for hits in ECL2, helix V and VI, whereas all ligands gave a similar pattern in the hinge region, helix I, II, III, VII and ECL3. In ECL2, agonists gave a distinct cluster of three hits ($K262^{ECL2}$, $P264^{ECL2}$, $G265^{ECL2}$) versus the four successive hits found for the antagonists ($F260^{ECL2}$, $G261^{ECL2}$, $K262^{ECL2}$, $R263^{ECL2}$). In helix V, the 33-mer antagonists featured intense hits ($Y272^{5.39}$, $Q273^{5.40}$ and $I277^{5.44}$) one helix turn deeper than the mapped agonist hits. Similarly, hits $I325^{6.51}$ and $T326^{6.52}$ in helix VI were visible only for the longer antagonists and are located one helix turn below the prominent hits $L329^{6.55}$ and $F330^{6.56}$ found with all ligands. Notably, three hits at the tip of helix V ($D269^{5.36}$, $Y270^{5.37}$ and $Y267^{ECL2}$, the latter two pointing out of the TMD in the 3D structure) (**Hollenstein et al., 2013**) were accessible for the two agonists, but were missing for all three antagonists.

## Pair-wise chemical crosslinking

We then focused on deciphering further details of the natural agonist CRF and the full antagonist [dFX]CRF(12-41) binding to CRF1R via pair-wise chemical crosslinking. We pinpointed intermolecular pairs of proximal ligand-receptor residues using the reaction between Cysteine (Cys) thiols incorporated into the CRF1R and mildly electrophilic α-chloroacetamide (ClAc) moieties incorporated into the peptide (**Figure 3A**), which occurs only when the two groups are proximal to each other in the associated complex (**Coin et al., 2013**; **Xiang et al., 2013, 2014**). Compared to classic disulphide

**Table 1.** Properties of synthesized [Lys(ClAc)]-peptide derivatives. X represents substitution of the two Met residues in position 21 and 38 to Norleucine. dF indicates the N-terminal D-Phe[12] of the CRF(12-41) analogues. $EC_{50}$ values of agonists were derived from the function of cAMP level in transiently transfected HEK293 cells and are shown as mean ± s.e.m. The percentage of receptor activation at 100 nM concentration of antagonists are normalized to either Ucn1 or CRF and are shown as mean ± s.e.m [%]. All values are obtained from at least three independent experiments, each performed in triplicate. Purity is given as the area% of the peak corresponding to the peptide with respect to total area in analytic HPLC (UV detection, 220 nm). Molecular weights and m/z values from MALDI-Tof mass spectrometry are monoisotopic.

| Peptides | Receptor activation | Analytic data | | |
|---|---|---|---|---|
| | | Purity (Area%) | M calculated | [M+H]+ found |
| **Agonists** | **$EC_{50}$ [nM]** | | | |
| Ucn1 | 0.16 ± 0.11 | | | |
| CRF | 0.11 ± 0.13 | | | |
| [Lys(ClAc)12]-XCRF | 3.39 ± 0.02 | >96% | 4775.57 | 4776.52 |
| [Lys(ClAc)13]- XCRF | 0.03 ± 0.16 | >96% | 4785.58 | 4786.55 |
| [Lys(ClAc)14]- XCRF | 0.79 ± 0.07 | >95% | 4809.56 | 4810.52 |
| [Lys(ClAc)15]- XCRF | 0.08 ± 0.22 | >95% | 4809.56 | 4810.55 |
| [Lys(ClAc)16]- XCRF | 0.20 ± 0.03 | >95% | 4766.54 | 4767.58 |
| [Lys(ClAc)17]- XCRF | 0.08 ± 0.26 | >95% | 4793.60 | 4794.64 |
| [Lys(ClAc)18]- XCRF | 0.03 ± 0.16 | >97% | 4823.57 | 4824.30 |
| [Lys(ClAc)31]- XCRF | 0.06 ± 0.11 | >98% | 4851.61 | 4852.88 |
| [Lys(ClAc)33]- XCRF | 0.04 ± 0.16 | >95% | 4835.61 | 4836.90 |
| **Antagonists** | **Receptor activation at 100 nM [%]** | | | |
| Ucn1(8-40) | 8.20 ± 2.76 | | | |
| CRF(9-41) | 4.03 ± 1.27 | | | |
| dFXCRF(12-41) | 0.53 ± 0.47 | | | |
| [ClAc0]- dFXCRF(12-41) | 0.62 ± 0.57 | >95% | 3612.97 | 3613.91 |
| [Lys(ClAc)13]- dFXCRF(12-41) | 2.41 ± 0.71 | >95% | 3604.00 | 3605.09 |
| [Lys(ClAc)14]- dFXCRF(12-41) | 1.49 ± 0.98 | >95% | 3627.98 | 3628.99 |
| [Lys(ClAc)15]- dFXCRF(12-41) | 0.39 ± 0.27 | >97% | 3627.98 | 3628.97 |
| [Lys(ClAc)16]- dFXCRF(12-41) | 1.06 ± 0.81 | >95% | 3584.96 | 3585.99 |
| [Lys(ClAc)17]- dFXCRF(12-41) | 0.31 ± 0.10 | >95% | 3612.02 | 3613.08 |
| [Lys(ClAc)18]- dFXCRF(12-41) | 0.59 ± 0.14 | >95% | 3641.99 | 3643.07 |
| [Lys(ClAc)31]- dFXCRF(12-41) | 0.52 ± 0.15 | >95% | 3670.02 | 3671.03 |
| [Lys(ClAc)33]- dFXCRF(12-41) | 1.85 ± 0.19 | >99% | 3654.03 | 3655.13 |

trapping (*Dong et al., 2016*; *Monaghan et al., 2008*), this strategy bypasses issues with self-dimerization of Cys-ligands and allows SDS-PAGE analysis under reducing conditions.

Cys residues were introduced into CRF1R sites corresponding to the 30 strongest hits of Azi crosslinking including both agonist and antagonist hits. In addition, seven positions in the ECD were rationally selected based on the 3D structures of the ligand-bound CRF1R ECD (*Grace et al., 2010*; *Pioszak et al., 2008*). Out of the 37 designed Cys-CRF1R mutants, 35 were expressed on the cell surface and retained the ability to bind CRF1R ligands, as demonstrated by photo-labeling with the Bpa[12]-Ucn1 ligand (*Kraetke et al., 2005*; *Coin et al., 2013*) (*Figure 3—figure supplement 1A*).

The ClAc moieties were installed into both CRF and dFXCRF(12-41), either as Lysine derivatives [Lys(ClAc)] or directly attached to the N-terminal α-amino group [ClAc0], for a total of nine ClAc-bearing analogues for each ligand (*Table 1*). Substituted positions cover all residues between V18, which is the putative entrance point of the ligands into the TMD pocket based on homology with our previous model (*Coin et al., 2013*), and position dF12, which is the N-terminal residue of the

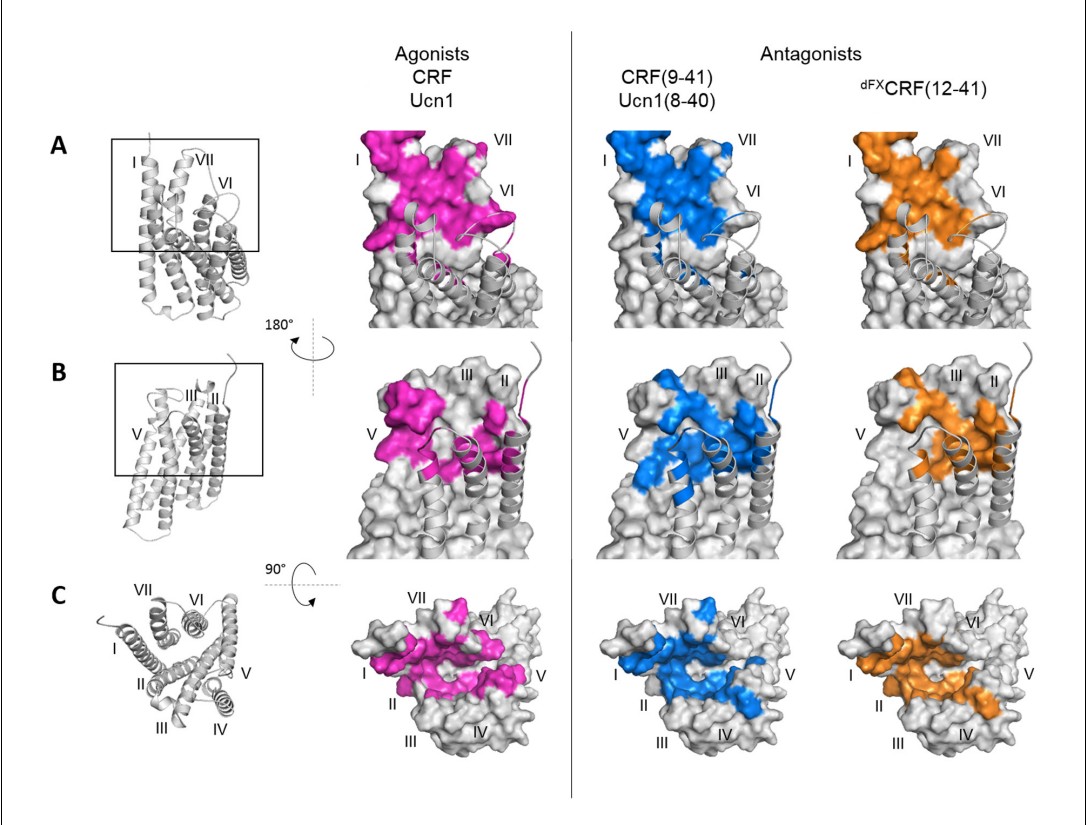

**Figure 2.** Footprints of peptide agonists and antagonists on CRF1R. Surface representation of the CRF1R TMD adapted from *Coin et al. (2013)*. The CRF1R ECD is not shown. Positions of CRF1R that crosslinked the ligand when substituted by Azi are highlighted. Footprints of the peptide agonists CRF and Ucn1 are highlighted in magenta and footprints of the antagonists CRF(9-41) and Ucn1(8-40) in blue. The footprint of the antagonist $^{dFX}$CRF (12-41) is highlighted in orange. (A) Side view of the binding pocket from the membrane plane showing the surface of helices I, VI and VII. Helices II, III, IV and V are drawn as cartoon. (B) Side view of the binding pocket from the membrane plane showing the surface of helices II, III, IV and V. Helices I, VI and VII are drawn as cartoon. (C) Top view into the binding pocket from the extracellular side.

The following figure supplement is available for figure 2:

**Figure supplement 1.** Snake plot of rat CRF1R with highlighted pair-wise and photo-crosslinking data obtained with (A) CRF, (B) $^{dFX}$CRF(12-41) and (C) Ucn1 (adapted from [*Coin et al., 2013*]).

antagonist. In addition, Lys(ClAc) was introduced at positions A31 and S33 in the mid region of the peptides, which are expected to interact with the hinge region and ECD of CRF1R. As demonstrated by cAMP accumulation assays, all agonists tolerated the introduced substitutions except for [Lys (ClAc)$^{12}$]-CRF, which was excluded from further experiments (*Table 1*). The Lys(ClAc) substitutions were well tolerated also by the antagonists, which behaved essentially like $^{dFX}$CRF(12-41) both in the cAMP accumulation assay and in photo-crosslinking experiments with Azi-CRF1R mutants (*Table 1* and *Figure 3—figure supplement 1B*).

Chemical crosslinking experiments between ClAc-ligands and Cys-receptors were performed in two blocks for a total of 373 combinations. Based on the two-domain binding model, ligands carrying the ClAc moiety toward the C-terminus were combined with CRF1R mutants bearing Cys in the ECD, whereas ligands substituted in the N-terminal domain were combined with the Cys-CRF1R set substituted in the TMD. Occurrence of pair-wise crosslinking was examined via immunoblotting using an anti-CRF antibody, as described above for the Azi photo-crosslinking experiments. We obtained distinct bands for a subset (5%) of ligand-receptor combinations (*Figure 3B*). The C-terminus of the agonist and the antagonist gave identical pair-wise hits with the CRF1R ECD (A31-E104$^{ECD}$ and S33-Y73$^{ECD}$). Likewise, both peptides featured the pair-wise hit V18-V120$^{1.40}$. However,

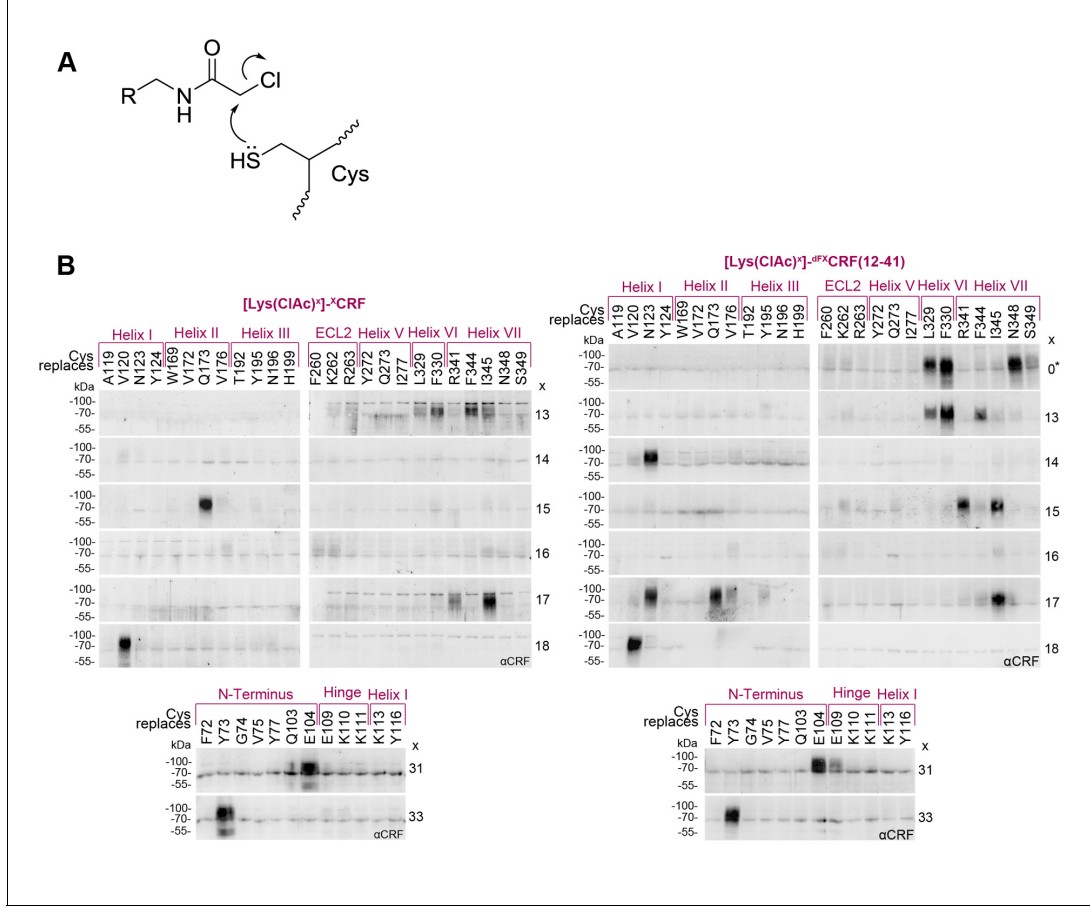

**Figure 3.** Cys-ClAc pair-wise crosslinking to pinpoint intermolecular pairs of proximal amino acids in ligand-CRF1R complexes. (**A**) Nucleophilic substitution reaction between a cysteine (Cys) thiol and a α-chloroacetamide (ClAc) moiety. (**B**) Transiently transfected 293T cells expressing each Cys-CRF1R mutant were incubated with each of the ClAc-peptide ligands. The mutation site in CRF1R is indicated in the upper row. The positions of ClAc moiety in the ligands are indicated in the right column of each panel. The peptide indicated with a star (*) bears the ClAc not on the side chain of a Lys, but directly on the N-terminus (see *Table 1*). Whole-cell lysates were separated on 10% SDS-PAGEs and analyzed by Western blotting using an anti-CRF antibody. The non-deglycosylated ligand-CRF1R complex runs at an apparent molecular weight of ~70–100 kDa (*Coin et al., 2013*). The non-crosslinked ligand is not detected (MW ~3–4 kDa). Signals were considered as hits if their intensity was higher than a threshold defined as 50% of the most intense signal per ligand.

The following figure supplement is available for figure 3:

**Figure supplement 1.** Photo-labeling of Cys-CRF1R mutants and photo-crosslinking of [Lys(ClAc)]-dFXCRF(12-41) analogues.

from E17 on toward the N-terminus, the two-dimensional crosslinking fingerprints of agonist and antagonist differ, showing different interaction sets with the receptor.

Overall, pair-wise crosslinking experiments revealed 7 proximal pairs of ligand-receptor amino acids for the CRF-CRF1R complex and 15 pairs for the dFXCRF(12-41)-CRF1R complex (*Table 2*). Based on the geometry of the Lys(ClAc) moiety, we estimate that the $C\beta$-$C\beta$ distance for the reaction between Lys(ClAc) and Cys to take place should not exceed 10 Å. For hits obtained with [ClAc0]-dFXCRF(12-41), which bears the ClAc moiety on the N-terminus, we estimate a maximal distance of 5 Å between the Nα of the peptide and the $C\beta$ of the receptor residue (*Figure 4A*). This geometrical estimation is further supported by Monte Carlo sampling experiments (*Figure 4B*).

## 3D models for agonist- and antagonist-bound CRF1R

Spatial constraints derived from pair-wise crosslinking were applied as soft harmonic distance restraints in energy-based conformational sampling to obtain 3D atomistic models of agonist- and

**Table 2.** C$\beta$-C$\beta$ and N-C$\beta$ interresidue distance restraints, measured in the molecular models of the CRF- and [dFX]CRF(12-41)-CRF1R complexes.

**CRF1R bound to agonist CRF**

| No | CRF-CRF1R residue pair | Region of CRF | Region of CRF1R | C$\beta$-C$\beta$ distance [Å] | |
|---|---|---|---|---|---|
| | | | | Constraint | Measured |
| 1 | H13-F330 | N-term | Helix VI | 10.0 | 5.1 |
| 2 | H13-F344 | N-term | Helix VII | 10.0 | 9.4 |
| 3 | L15-Q173 | N-term | Helix II | 10.0 | 7.1 |
| 4 | E17-I345 | N-term | Helix VII | 10.0 | 9.2 |
| 5 | V18-V120 | C-term | Helix I | 10.0 | 6.4 |
| 6 | A31-E104 | C-term | Hinge | 10.0 | 8.6 |
| 7 | S33-Y73 | C-term | ECD | 10.0 | 6.7 |

**CRF1R bound to Antagonist [dFX]CRF(12-41)**

| No | CRF1R-[dFX]CRF(12-41) residue pair | Region of [dFX]CRF(12-41) | Region of CRF1R | C$\beta$-C$\beta$ or N-C$\beta$* distance [Å] | |
|---|---|---|---|---|---|
| | | | | Constraint | Measured |
| 1 | F12-L329 | N-term | Helix VI | 5.0* | 4.4* |
| 2 | F12-F330 | N-term | Helix VI | 5.0* | 4.6* |
| 3 | F12-N348 | N-term | Helix VII | 5.0* | 4.8* |
| 4 | H13-L329 | N-term | Helix VI | 10.0 | 6.7 |
| 5 | H13-F330 | N-term | Helix VI | 10.0 | 5.5 |
| 6 | H13-F344 | N-term | Helix | 10.0 | 5.0 |
| 7 | L14-N123 | N-term | Helix I | 10.0 | 9.1 |
| 8 | L15-R341 | N-term | Helix VII | 10.0 | 5.9 |
| 9 | L15-I345 | N-term | Helix VII | 10.0 | 6.8 |
| 10 | E17-N123 | N-term | Helix I | 10.0 | 7.4 |
| 11 | E17-Q173 | N-term | Helix II | 10.0 | 6.3 |
| 12 | E17-I345 | N-term | Helix VII | 10.0 | 6.1 |
| 13 | V18-V120 | C-term | Helix I | 10.0 | 4.7 |
| 14 | A31-E104 | C-term | Hinge | 10.0 | 4.3 |
| 15 | S33-Y73 | C-term | ECD | 10.0 | 6.7 |

*Pair-wise crosslinking between N-terminal ClAc in the peptide and Cys thiol in CRF1R.

antagonist-bound CRF1R. CRF and [dFX]CRF(12-41) were initially docked into a flexible model of full-length rat CRF1R. The model represents the conformation of Ucn1-bound CRF1R from our previous study (*Coin et al., 2013*) and was derived from the combination of crystal structures of human CRF1R ECD (PDB: 3EHU) (*Pioszak et al., 2008*) and of the thermostabilized human CRF1R TMD (PDB: 4K5Y) (*Hollenstein et al., 2013*).

Extensive sampling of peptide and receptor conformations converged to optimized models for the CRF-CRF1R and the [dFX]CRF(12-41)-CRF1R complexes (*Figure 5*) where all 7 or 15 experimentally derived pair-wise restraints, respectively, were satisfied (*Table 2*). Both models are further validated by Azi-crosslinking hits, which have not been used as restraints in the simulations (*Figure 5—figure supplement 1*). All 56 photo-crosslinking hits, with the only exception of E336[ECL3] in the highly flexible ECL3 (see 'Molecular dynamics' section below), were found within the estimated effective radius of Azi crosslinking, with distances between the C$\beta$ of the crosslinking residues and the nearest non-hydrogen atom of the ligand not exceeding 9 Å (*Figure 4C*) (*Coin et al., 2013*).

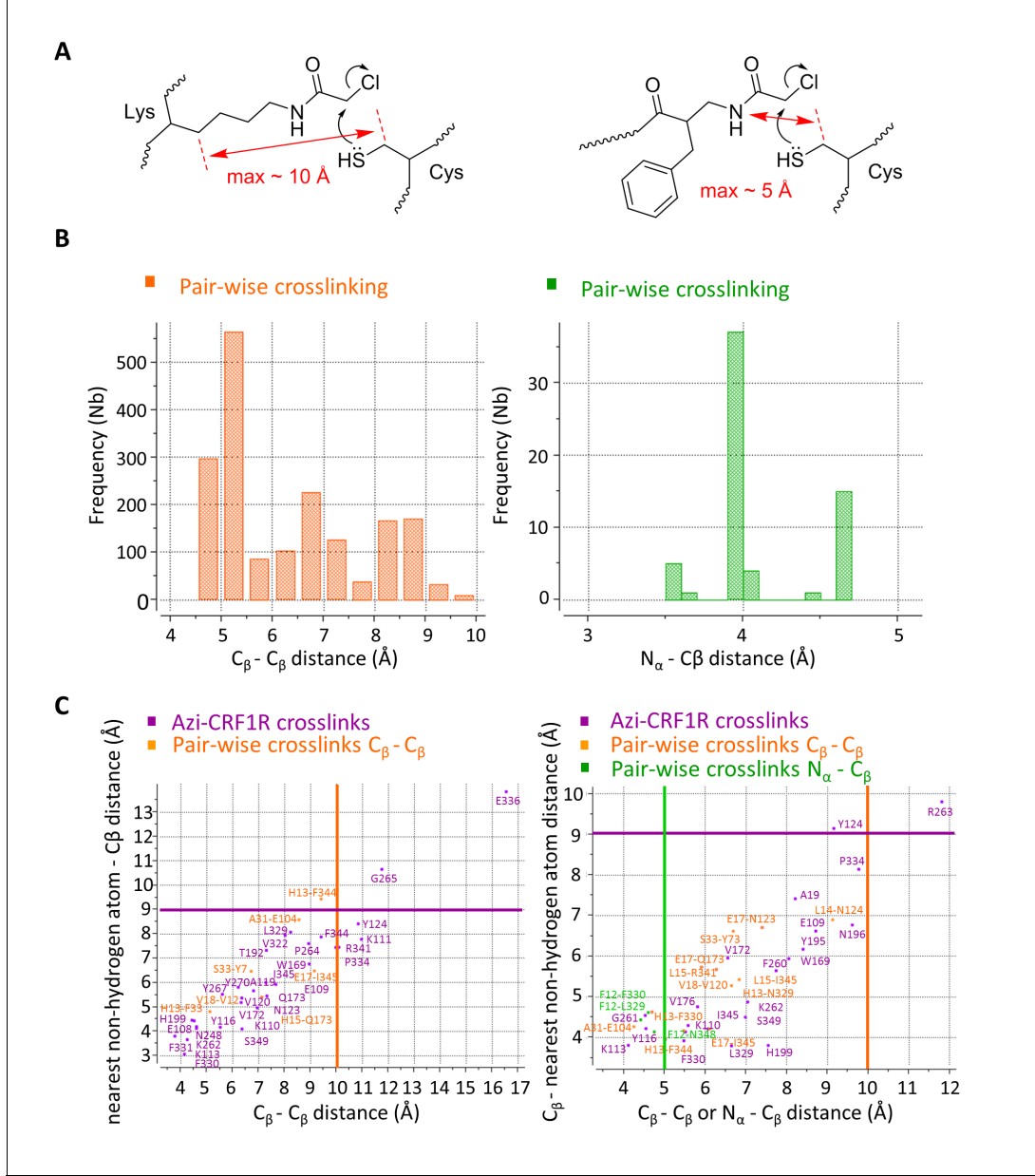

**Figure 4.** Atom-atom distances of residue pairs in the peptide-CRF1R complexes. (**A**) Geometrical estimation of the Cβ-Cβ distance or N-Cβ distance for the ClAc-Cys reaction to happen when the ClAc moiety is installed on the side chain of a Lysine residue (left) or at the N-terminus of the peptide (right). (**B**) Range of sterically possible Cβ-Cβ distances (left) or N-Cβ distances (right) in the covalent crosslinked product for residues involved in pair-wise crosslinking. The distance range distributions were calculated based on the Monte Carlo sampling of the free molecule conformations in ICM software. (**C**) Measured distances for Azi photo-crosslinking and pair-wise chemical crosslinking hits between ligand and receptor in the predicted model of the CRF-CRF1R (left) and the $^{dFX}$CRF(12-41)-CRF1R complex (right). For Azi photo-crosslinking hits (magenta), distances are measured from the Cβ atom of the indicated CRF1R residue to the nearest Cβ atom of CRF (x-axis) or to the nearest non-hydrogen atom of CRF (y-axis) or $^{dFX}$CRF(12-41). For pairs of amino acids involved in chemical crosslinking, distances plotted along the x-axis are measured either as Cβ-Cβ (orange) or N-Cβ (green). The horizontal magenta line shows the approximate 9 Å radius for the reach of Azi crosslinking (from Cβ of Azi), vertical lines show the approximate 10 Å (orange) and 5 Å (green) cutoff for pair-wise crosslinking between Cβ-Cβ or N-Cβ, respectively.

## Agonist-bound CRF1R model

Docked CRF forms a long α-helical segment from the C-terminus to residue V18 (*Figure 5*). The C-terminal segment (I41-V18) binds to the CRF1R ECD similarly to the crystal structure of CRF1R ECD bound to CRF(26-41) (PDB: 3EHU) (*Pioszak et al., 2008*), and approaches the TMD antiparallel

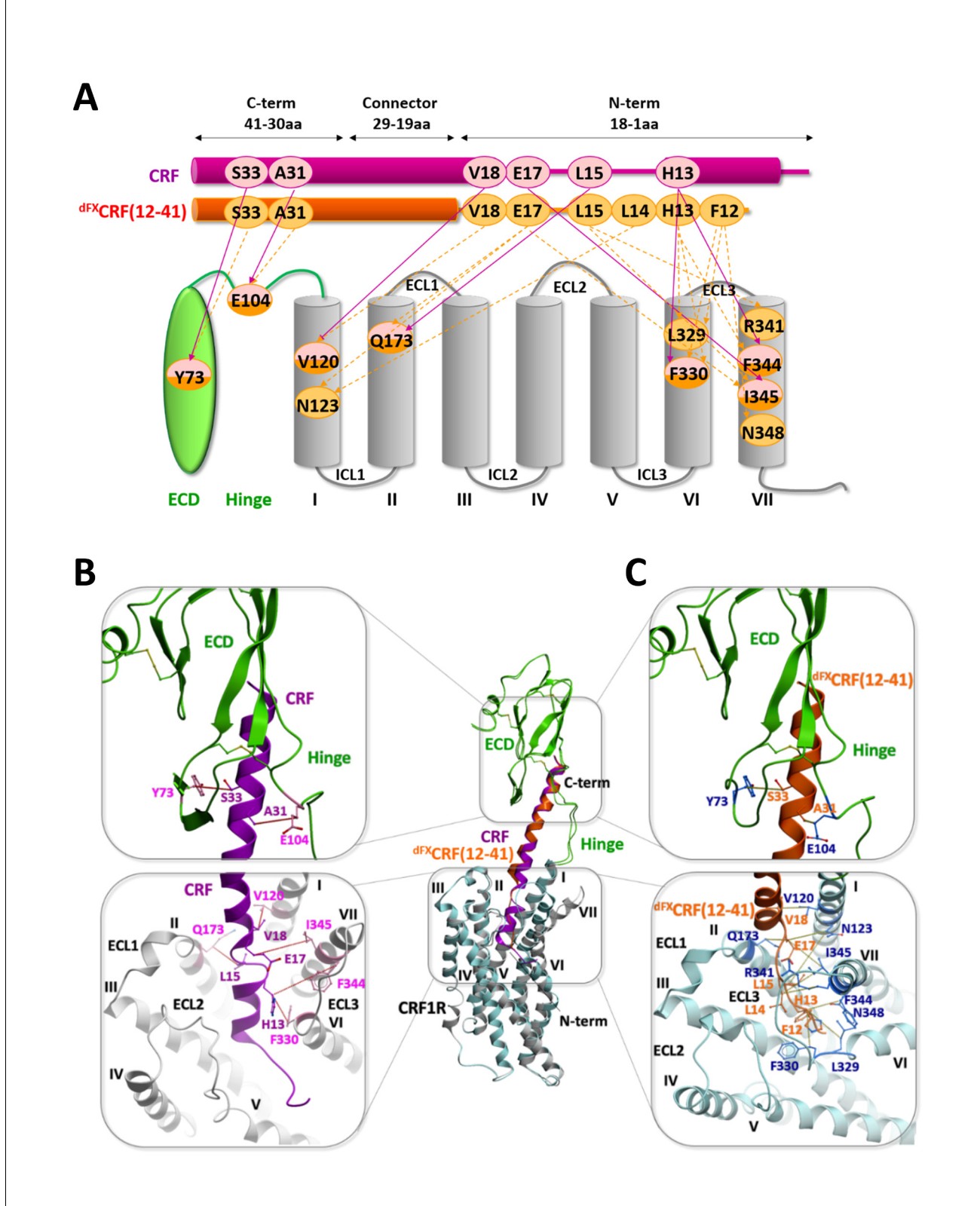

**Figure 5.** Comparison of the agonist-bound model CRF-CRF1R with the antagonist-bound model $^{dFX}$CRF(12-41)-CRF1R. (A) Pair-wise crosslinking hits of CRF (magenta) and $^{dFX}$CRF(12-41) (orange). (B, C) Overall view and zoom into the details of the superimposed models of the CRF (magenta)-CRF1R (grey) complex and the $^{dFX}$CRF(12-41) (orange)-CRF1R (light blue) complex. The framed sections are magnified in the panels (B) (CRF-CRF1R) and (C)
*Figure 5 continued on next page*

Figure 5 continued

(dFXCRF(12-41)-CRF1R). Residue pairs shown in sticks and connected by dotted lines (distances are presented in *Table 2*) indicate distance restraints derived from pair-wise crosslinking.

The following figure supplements are available for figure 5:

**Figure supplement 1.** Validation of the peptide-CRF1R complexes by photo-crosslinking hits.

**Figure supplement 2.** Comparison of the two agonist-bound models Ucn1-CRF1R and CRF-CRF1R.

to the helical stalk of helix I. As CRF enters the TMD in the groove formed by the helices I, II and VII, experimental constraints L15-Q173$^{2.68}$, H13-F330$^{6.56}$ and H13-F344$^{7.38}$ impose disruption of the α-helical conformation and the formation of a short loop in the segment R16-L14. Following this loop, docked CRF shows a second α-helical segment (H13-S7) that lies across the TMD interface. The ligand reaches six of the seven TM helices as deep as Y124$^{1.44}$, W169$^{2.64}$, H199$^{3.40}$, Q273$^{5.40}$, L329$^{6.55}$ and S349$^{7.43}$ (*Figure 1—figure supplement 1*). No interactions are predicted by the model with helix IV, in accord with the absence of any experimental crosslinking in this helix. Finally, the very N-terminus of CRF (I6-S1) exits the binding pocket through a gap between helix V and VI and extends beyond the TMD.

Overall, the conformation of the docked CRF agonist is similar to the conformation adopted by Ucn1 in the Ucn1-CRF1R model in our previous study (*Coin et al., 2013*), even though the two models are based on distinct and independent sets of experimental constraints. Notably, the Ucn1 model predicted the kink in the α-helical fold of the agonist at the entrance of the TMD pocket even without pair-wise crosslinks in this region, while new experimental constraints further enforce this conformation in CRF (*Figure 5—figure supplement 2*).

Both agonist models predict a modest 3.1 Å inward shift of the extracellular tip of helix VII and a movement of ECL3, which compacts the orthosteric binding pocket around the peptide with respect to the wide-open conformation observed in the crystal structure of the CRF1R TMD bound to the small molecule antagonist CP-376395 (PDB: 4K5Y) (*Hollenstein et al., 2013*). This compaction results from receptor flexibility at the 'backbone hinges' Q355$^{7.49}$ and G356$^{7.50}$ and at ECL3, leads to an overall gain in the conformational energy of the complex and helps to satisfy the pair-wise distance restraints. Importantly, the TMD model of the CRF-CRF1R complex, based only on restraints identified in this study, satisfies the Ucn1 restraints in the Ucn1-CRF1R model and vice versa, so that the two models strongly validate each other (*Table 3*). The only substantial difference between CRF and Ucn1-bound CRF1R models is found in the ECD domain, which is now rotated and shifted about 5 Å. This adjustment in the ECD position was induced by the newly derived pair-wise constraints in the C-terminal segment of the peptide, which were missing in the Unc1-CRF1R model, and helped to optimize the key interactions of CRF with the hinge region and ECD pocket.

## Antagonist-bound CRF1R model

Docked dFXCRF(12-41) antagonist shows both similarities and major differences compared to the conformation of CRF agonist (*Figure 5*). The C-terminal segments (I41-V18) of both agonist and antagonist similarly interact with CRF1R ECD, the hinge region and the helical stalk of helix I, which accounts for identical patterns of pair-wise restraints and Azi-footprints in these regions (*Figures 1– 3*, *Figure 5—figure supplement 1*). However, at the entrance of the binding pocket, where residue E17 of CRF gave chemical crosslinking only with helix VII, the E17 of dFXCRF(12-41) crosslinked to helix VII (I345$^{7.39}$), helix I (N123$^{1.43}$), and helix II (Q173$^{2.68}$). Accordingly, the restrained docking placed E17 about equidistant from the three helices, with the side chain pointing toward the bottom of the TMD. As a consequence, while the C-terminal segment of CRF agonist kept the α-helical conformation up to E17 inclusively, the α-helix in the dFXCRF(12-41) antagonist already unravelled at residue E17. This conformational change is further supported by an additional weak signal of E17-V176$^{2.71}$ pair-wise crosslinking visible for the antagonist (*Figure 3*), which was not considered for modeling, but satisfies the 10 Å Cβ-Cβ distance in the model. From E17, docked dFXCRF(12-41) keeps the non-helical conformation all the way up to its N-terminal residue (dF12), which accounts

**Table 3.** (A) Cβ-Cβ Interresidue distance restraints experimentally derived for the CRF-CRF1R complex, measured in the model of the CRF-CRF1R complex, and measured at homologous positions in the model of the Ucn1-CRF1R complex. (B) Cβ-Cβ Interresidue distance restrains experimentally derived for the Ucn1-CRF1R complex, measured in the model of the Ucn1-CRF1R complex, and measured at homologous positions in the model of the CRF-CRF1R complex.

| A | | Agonist CRF | | | | Agonist Ucn1 | |
|---|---|---|---|---|---|---|---|
| No | CRF-CRF1R residue pair | Position in the ligand | Position in CRF1R | Cβ-Cβ [Å] constraint | Cβ-Cβ [Å] in the CRF model | Cβ-Cβ [Å] in the Ucn1 model | |
| 1 | **H13-F330** | **N-term** | **Helix VI** | **10.0** | **5.1** | H12-F330 | 5.3 |
| 2 | H13-F344 | N-term | Helix VII | 10.0 | 9.4 | H12-F344 | 8.7 |
| 3 | L15-Q173 | N-term | Helix II | 10.0 | 7.1 | L14-Q173 | 6.5 |
| 4 | E17-I345 | N-term | Helix VII | 10.0 | 9.2 | T16-I345 | 6.0 |
| 5 | V18-V120 | C-term | Helix I | 10.0 | 6.4 | L17-V120 | 4.1 |
| 6 | A31-E104 | C-term | Hinge | 10.0 | 8.6 | A30-E104 | 12.3 |
| 7 | S33-Y73 | C-term | ECD | 10.0 | 6.7 | Q32-Y73 | 10.1 |
| B | | Agonist Ucn1 | | | | Agonist CRF | |
| No | Ucn1-CRF1R Residue Pair | Position in the ligand | Position in CRF1R | Cβ-Cβ [Å] Constraint | Cβ-Cβ [Å] In the Ucn1 model | Cβ-Cβ [Å] In the CRF model | |
| 1 | D8-Q273 | N-term | Helix V | 9.0 | 8.6 | D9-Q273 | 9.0 |
| 2 | D8-F330 | N-term | Helix VI | 9.0 | 4.5 | D9-F330 | 4.2 |
| 3 | H12-L329 | N-term | Helix VI | 9.0 | 8.0 | H13-L329 | 8.5 |
| 4 | **H12-F330** | **N-term** | **Helix VI** | **9.0** | **5.3** | H13-F330 | 5.1 |
| 5 | H12-N333 | N-term | Helix VI | 9.0 | 6.9 | H13-N333 | 6.1 |
| 6 | H12-I345 | N-term | Helix VII | 9.0 | 8.4 | H13-I345 | 9.3 |
| 7 | H12-N348 | N-term | Helix VII | 9.0 | 6.5 | H13-N348 | 6.8 |
| 8 | H12-S349 | N-term | Helix VII | 9.0 | 9.4 | H13-S349 | 10.7 |
| 9 | L14-S349 | N-term | Helix VII | 9.0 | 9.0 | L15-S349 | 9.6 |

Bold – residue pairs tested in both models.

Blue – residue pairs that satisfied distance restraints.

Red – residue pairs that did not exactly satisfiy distance restraints.

for the series of antagonist-specific pair-wise crosslinking in this region, including crosslinking of L15 with R341[7.35] and I345[7.39], crosslinking of L14 with N123[1.43] and crosslinking of H13 with L329[6.55], F330[6.56] and F344[7.38]. Finally, while the N-terminal segment of the CRF agonist is solvent exposed outside the pocket, the N-terminus (dF12) of the [dFX]CRF(12-41) antagonist is buried inside the TMD, which is corroborated by chemical crosslinking of dF12 with L329[6.55], F330[6.56] and N348[7.42] (*Figure 3*).

Compared to both the CRF1R crystal structure and the agonist-bound complexes, the predicted conformation of the TMD orthosteric pocket in the [dFX]CRF(12-41)-CRF1R model is much more compact (*Figure 6*). Backbone flexibility at the 'backbone hinges' Q355[7.49] and G356[7.50] and extensive conformational sampling of the TMD helical bundle in the modeling procedure led to a dramatic inward shift of the extracellular tips of helices VI and VII by ~6 Å and ~9 Å, respectively, as compared to the conformation of the small-molecule bound TMD crystal (PDB: 4K5Y) (*Hollenstein et al., 2013*) (*Figure 6A*, red arrows indicate the inward shift) . This shift of the helices was also accompanied by a shift in the ECL3, which formed a 'lid' on top of the N-terminal segment of the bound antagonist. As a control, we attempted docking of the [dFX]CRF(12-41) peptide in the rigid TMD model that maintains a 'wide' conformation of the receptor as in the CRF-CRF1R agonist model. In this case, interactions of the linear N-terminal segment (E17-dF12) of the antagonist in the wide pocket were less energetically favourable. Moreover, the wide conformation of the pocket and the extended linear folding of the antagonist would preclude formation of Azi photo-crosslinking on both helix VI-VII and helix III at the opposite sides of the pocket, which were observed

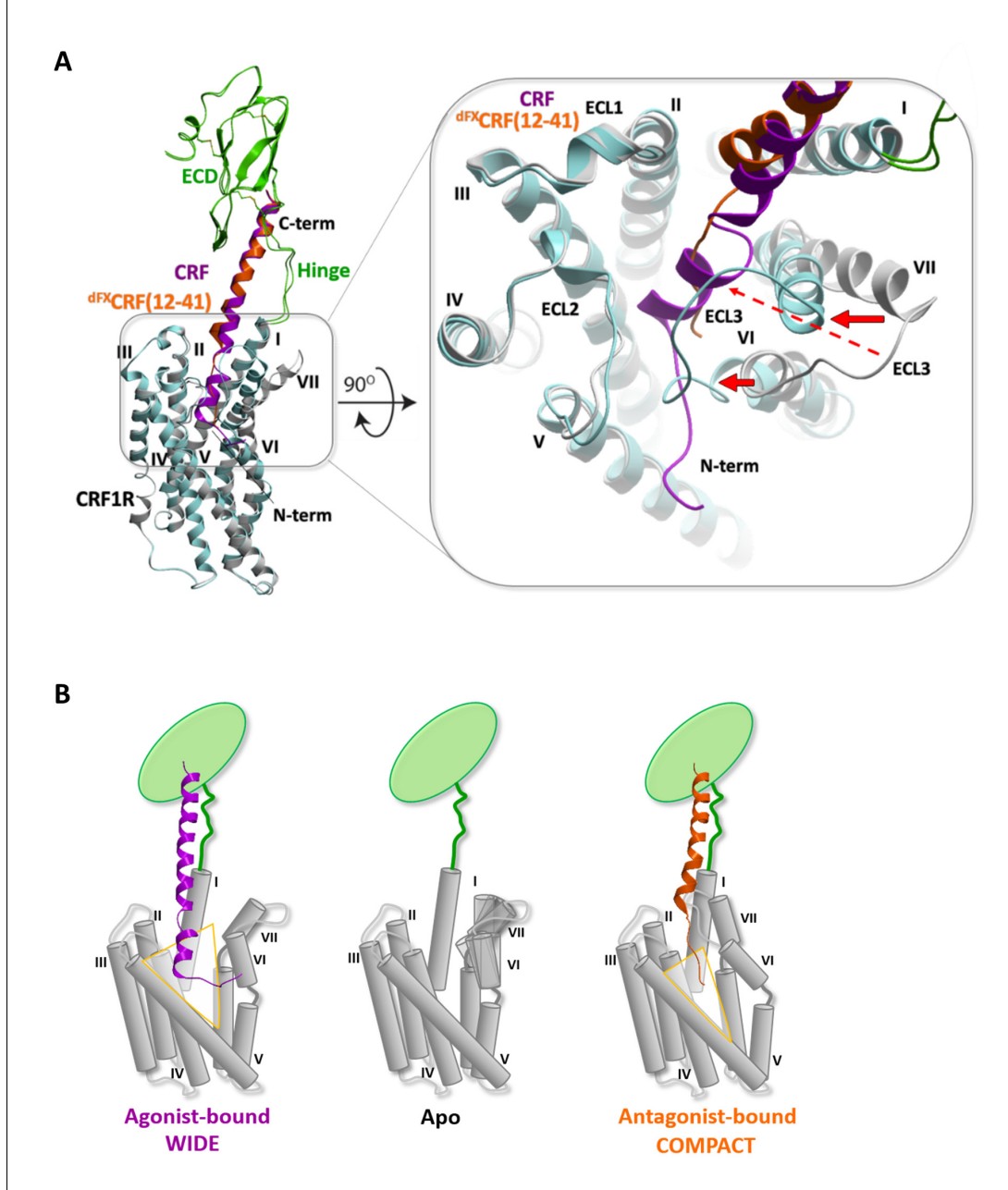

**Figure 6.** Conformational changes within the receptor between the agonist- and antagonist bound complexes. (**A**) Overall view and extracellular zoom into the binding pocket of the superimposed models CRF (magenta)-CRF1R (grey) and $^{dFX}$CRF(12-41) (orange)-CRF1R (light blue) highlighting the inward shift of helices VI and VII in the antagonist-bound model. (**B**) Schematic presentation of the active CRF (magenta)-CRF1R complex and the inactive $^{dFX}$CRF(12-41) (orange)-CRF1R complex showing the predicted conformational changes between the 'wide' agonist-bound and the 'compact' antagonist-bound states of the receptor. Apo CRF1R is expected to have a more flexible conformation of the TMD and sample a range of conformations.

experimentally (*Figure 1E*). Therefore, conformational changes of the TMD predicted in the flexible $^{dFX}$CRF(12-41)-CRF1R model are critical for explaining the Azi photo-crosslinking results, which were employed to validate the model. Overall, our models predict two distinct conformations for the TMD of CRF1R when bound to either agonist or antagonist, which we define as 'wide' and 'compact', respectively (*Figure 6*).

## Molecular dynamics

In order to investigate stability and flexibility of the agonist- and antagonist-bound models, we performed molecular dynamics (MD) simulations of the lipid-embedded peptide-receptor complexes for a period of 1 μs with or without applying the distance restraints derived from pair-wise crosslinking.

The CRF1R complex with CRF agonist maintained integrity in all restrained and unrestrained runs, with all seven amino acid pairs involved in chemical crosslinking retaining a close proximity throughout the simulation (*Figure 7A*). In unrestrained runs, some residue pairs (e.g. H13-F344[7.38]) dynamically fluctuated in and out of the range of 10 Å, which is compatible with the crosslinking reaction.

In these initial runs for the agonist-bound complex, we did not observe dramatic conformational rearrangements of helices in the intracellular part of the CRF1R, as a part of the known GPCR activation mechanism (*Rasmussen et al., 2011*; *Katritch et al., 2014*). Indeed, intracellular changes are usually softly coupled with extracellular changes and transitions may require much longer time scales than the 1 μs MD simulations here (*Nygaard et al., 2013*). To glean intracellular changes accompanying the activation of CRF1R, we therefore used the recent cryo-EM structure of the class B calcitonin receptor in complex with Gs-protein heterotrimer (PDB: 5UZ7) (*Liang et al., 2017*). To model the conformational change in CRF1R, we applied an outward shift of helix VI to the CRF-CRF1R model, as observed in the calcitonin receptor structure, and submitted it for the MD simulations. This conformation of the complex maintained integrity and ligand contacts in both restrained and unrestrained 1 μs runs, while also maintaining the outward position of helix VI as an indicator of active-like state (*Figure 8*).

The CRF1R complex with [dFX]CRF(12-41) antagonist also maintained overall integrity in both restrained and unrestrained simulations (*Figure 7B,C*), with most of the 15 distances determined with pair-wise crosslinking being retained within 10–12 Å. In some unrestrained runs, we observed up to 14 Å deviations in distances H13-F344[7.38] and L14-N123[1.43], and in contacts of dF12 with L329[6.55], F330[6.56] and N348[7.42]. Thus, simulations showed higher flexibility for the linear N-terminal segment of the antagonist in respect to the homologous α-helical segment of the agonist. As a control, we also conducted a series of MD simulations of the antagonist-bound CRF1R starting with the TMD in the agonist-bound 'wide' conformation. In this case, the receptor contacts with [dFX]CRF(12-41) were maintained in the ECD, but most pair-wise crosslinking distances quickly drifted far beyond 14 Å in the TMD, where most interactions with the antagonist N-terminus were completely lost. Therefore, MD showed that the 'wide' TMD conformation is less optimal than the 'compact' conformation for CRF1R interactions with the [dFX]CRF(12-41) antagonist (*Figure 7—figure supplement 1*).

Interestingly, while preserving inter-residue contacts, both the agonist- and the antagonist-bound models showed overall flexibility throughout the simulations. Especially variable was the orientation of the ECD relative to the TMD, which swayed as much as 15–20 Å relative to the membrane normal (*Figure 9A,B*). High flexibility in MD simulations was also observed for ECL3. Residue E336[ECL3], located about 14 Å away from the ligand in the CRF-CRF1R model, came as close as ~6 Å to the peptide during 800 ns simulation runs, thus explaining occurrence of Azi photo-crosslinking at this position (*Figure 1*, *Figure 9C*). High flexibility of ECL3 is also supported by the crystal structure of the CRF1R TMD, where only one of the three molecules composing the crystallographic unit shows well defined electron density for ECL3 (*Hollenstein et al., 2013*).

## Discussion

It has been largely believed that N-truncated peptide ligands of class B GPCRs behave as antagonists because they occupy the binding site in the ECD, but are too short to reach the TMD, where activation takes place (*Cordomi et al., 2017*). Our crosslinking results provide direct evidence that N-truncated antagonists of CRF1R extensively penetrate the TMD, which explains previous observations obtained with chimeric constructs and isolated receptor domains (*Hoare et al., 2004*, *2005*), as well as the fact that the 30-mer antagonist Astressin induces CRF1R internalization (*Perry et al., 2005*).

Compared to the natural agonists CRF and Ucn1, however, CRF1R antagonists give a different footprint on the receptor TMD, revealing distinct interactions with helix V, helix VI and with ECL2. While agonists protrude into a groove between helices V and VI, the 33-mer peptides, which behave as weak partial agonists at high concentration, do reach both helices (Azi-hits Y272[5.39], Q273[5.40],

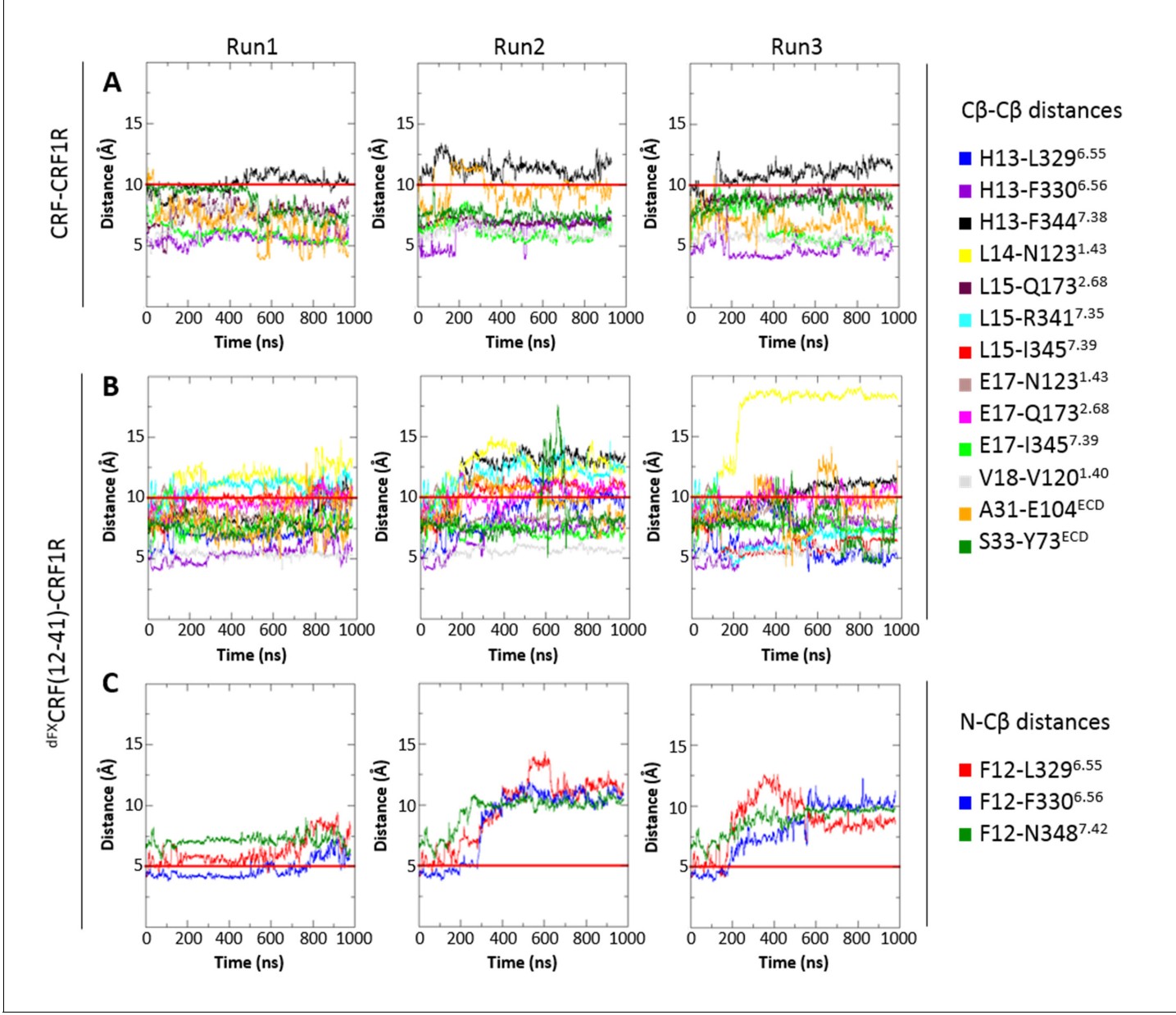

**Figure 7.** Interresidue distances during MD simulations. (**A**) Cβ-Cβ distances of CRF-CRF1R residue pairs during the MD simulations of the CRF-CRF1R complex. (**B, C**) Cβ-Cβ (**B**) or N-Cβ (**C**) distances of dFXCRF(12-41)-CRF1R residue pairs during the MD simulations of the dFXCRF(12-41)-CRF1R complex. MD Run1 was performed under 7 (**A**) or 15 (**B, C**) harmonic distance restraints corresponding to the crosslinked residue pairs (see *Table 2*). In MD Run2 and Run3, the distance restraints were removed after 20 ns in each run. The red horizontal line represents the 10 Å (**A, B**) or 5 Å (**C**) distance threshold.

The following figure supplement is available for figure 7:

**Figure supplement 1.** Interresidue distances during control MD simulations of antagonist dFXCRF(12-41)-wide-CRF1R complex.

I277[5.44], I325[6.51]) but do not penetrate through them (lack of crosslinking at Y267[ECL2], D269[5.36], Y270[5.37]). The 30-mer antagonist dFXCRF(12-41), which is not able to activate the receptor at all, shows no contact with helix V, although it maintains contacts with other helices. Taken together, these findings support our previous hypothesis that only agonists can push apart helices V and VI, which constitutes a part of the activation mechanism (*Coin et al., 2013*). Distinct agonist and antagonist footprints suggest also a role for ECL2 in CRF1R activation. Indeed, ECL2 is known for

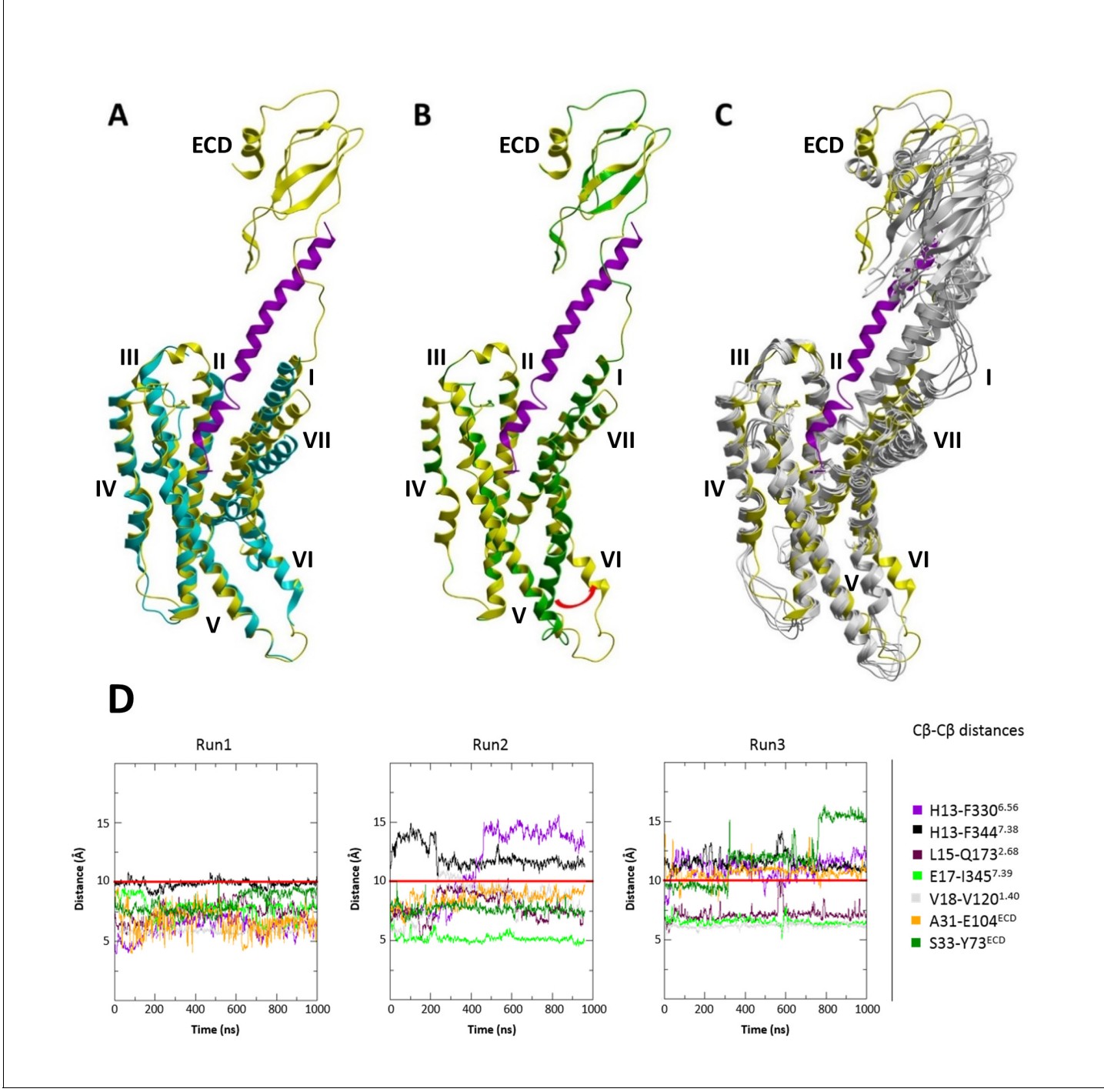

**Figure 8.** MD simulations of the modified CRF-CRF1R complex model with active-like conformation of the intracellular TMD region. (**A**) Superimposition of the cryo-EM structure of calcitonin receptor (PDB ID: 5UZ7) (cyan) and the modified active-like CRF (magenta)-CRF1R (yellow) complex model with outward movement of helix VI. (**B**) Superimposition of the initial CRF (magenta)-CRF1R (green) model and active-like CRF (magenta)-CRF1R (yellow). The red arrow indicates the outward movement of the intracellular part of helix VI in the active-like CRF-CRF1R complex. (**C**) Superimposition of the active-like CRF (magenta)-CRF1R (yellow) model with MD simulation snapshots obtained at 200 ns, 400 ns, 600 ns, 800 ns and 1 μs (gray) of MD simulation (Run1). (**D**) Interresidue distances during MD simulations. C$\beta$-C$\beta$ distances of CRF-CRF1R residue pairs during the MD simulations of the active-like CRF-CRF1R complex. MD Run1 was performed under 7 harmonic distance restraints corresponding to the crosslinked residue pairs (see *Table 2*). In MD simulation Run2 and Run3, the distance restraints were removed after 20 ns in each run. The red horizontal line represents the 10 Å distance threshold.

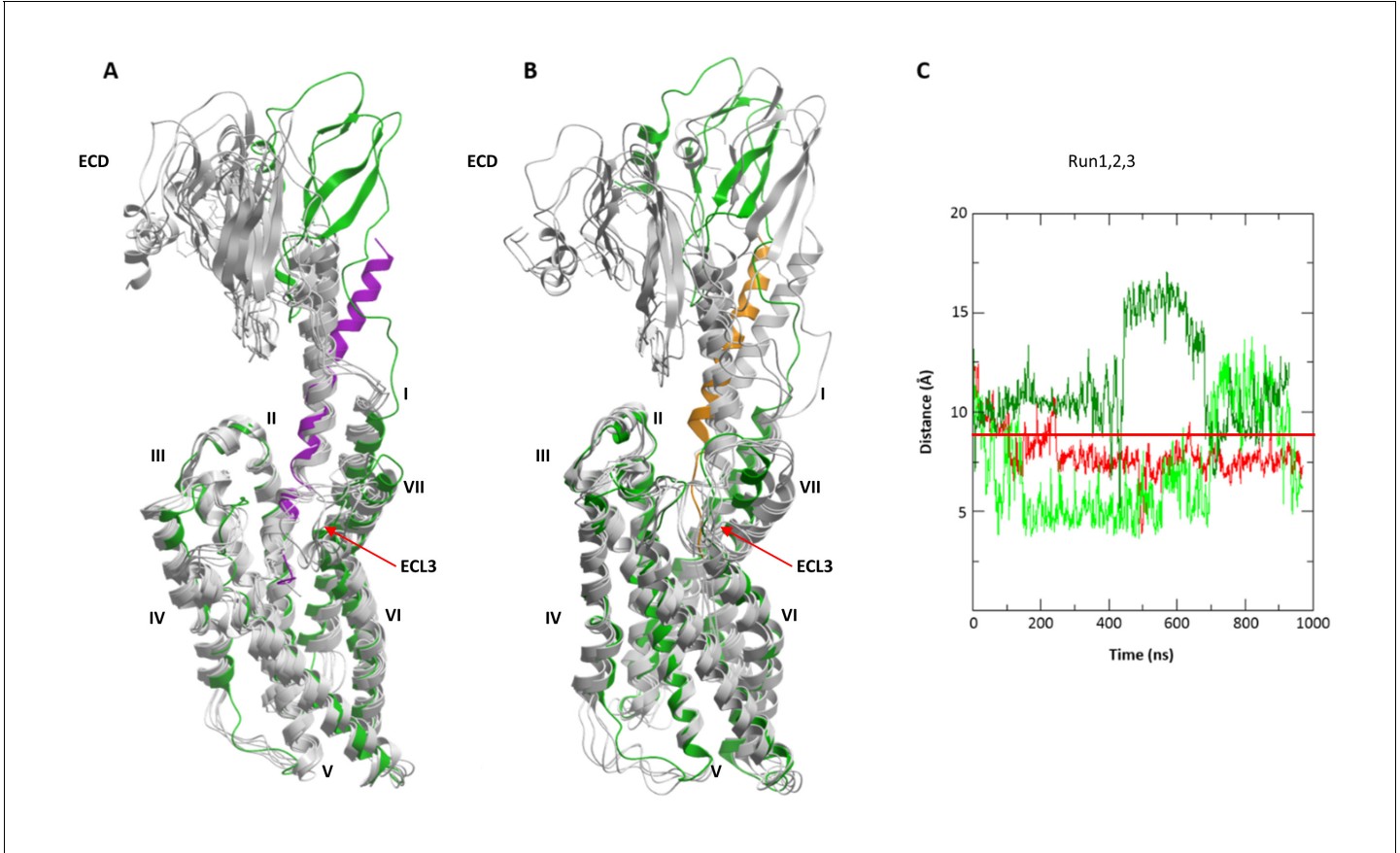

**Figure 9.** Conformational flexibility of ECD and ECL3 during MD simulations. (**A, B**) The range of ECD movement. (**A**) Superimposition of the initial CRF (magenta)-CRF1R (green) model with models obtained at 200 ns, 400 ns, 600 ns, 800 ns and 1 μs (gray) of unrestrained MD simulation (Run2). (**B**) Superimposition of the initial $^{dFX}$CRF(12-41) (orange)-CRF1R (green) model with models obtained at 100 ns, 200 ns, 300 ns, 400 ns and 500 ns (gray) of unrestrained MD simulation (Run2). (**C**) Conformational flexibility of ECL3 around residue E336$^{ECL3}$ in MD simulations of agonist CRF-CRF1R complex. Traces show minimum distances between the C$\beta$ atom of E336 and any heavy atom of CFR in Run1 (red, with restraints), Run2 (green, no restraints) and Run3 (light green, no restraints). The red horizontal line represents the 9 Å distance threshold.

contributing to ligand binding in CRF1R (*Assil-Kishawi and Abou-Samra, 2002*; *Gkountelias et al., 2009*), and a number of studies suggested a role of ECL2 in activation of class B GPCRs (*Koole et al., 2012*) and other GPCRs (reviewed in [*Wheatley et al., 2012*]).

Comparison of the atomistic models of CRF1R bound either to CRF or $^{dFX}$CRF(12-41) shows a different folding for the agonist and the antagonist in the CRF1R TMD, as well as different conformations of the receptor. Following a common C-terminal helical segment interacting with the CRF1R ECD and hinge region, both agonist and antagonist bend at the entrance of the TMD and place the N-terminal segment across the pocket almost parallel to the membrane plane. However, while CRF resumes the helical folding from H13 on, the N-terminus of $^{dFX}$CRF(12-41) keeps an extended conformation. This conformational difference is clearly required to satisfy the distinct patterns of pairwise crosslinks in the E17-F12 region of the peptides. The linear N-terminus of the antagonist shows higher flexibility in MD simulations as compared to the corresponding α-helical region of agonist, which suggests weaker and less defined interactions with the receptor and explains the ~10 fold lower binding affinity of CRF1R to $^{dFX}$CRF(12-41) than to CRF (*Gulyas et al., 1995*).

To account for the different conformations of the ligands, the energy-optimized models predict a major inward shift of the extracellular halves of helices VI and VII, which is independently validated by a set of photo-crosslinks (*Figure 1E*, *Figure 5—figure supplement 1*). Thus, CRF1R TMD compacts around the unstructured N-terminal segment of the antagonist, while it adopts a more open conformation to accommodate the bulkier α-helical segment (S7-H13) and the N-terminal tip (S1-I6)

of the agonist, which protrudes between helices V and VI beyond the pocket (*Figure 6B*). Apparently, stabilization of such a wide open TMD conformation is important for the activation of CRF1R by peptide agonists. Interestingly, the crystal structure of CRF1R TMD bound to the allosteric antagonist CP-376395 (PDB: 4K5Y) (*Hollenstein et al., 2013*) features an even more open V-shape. However, CP-376395 sits deep in the TMD closer to the G-protein-binding site and likely blocks the signal transmission via a mechanism that is distinct from the action of orthosteric peptide antagonists (*Hollenstein et al., 2014*).

Movements of the extracellular parts of helices VI and VII rely on Glycine hinges G324$^{6.50}$ and G356$^{7.50}$, which are fully conserved among class B GPCRs. Therefore, hinge flexibility in the apo state and stabilization of a specific conformation of these helices upon agonist binding (*Figure 6*) can be relevant also for the activation mechanism of other class B receptors. Specific details can differ for class B ligands that are shorter than CRF, such as glucagon (*Siu et al., 2013*; *Yang et al., 2015*), secretin (*Dong et al., 2016*) and GLP-1 (*Miller et al., 2011*; *Kirkpatrick et al., 2012*), which are thought to penetrate the TMD perpendicularly to the membrane plane. This mechanism may be even more distinct from class A GPCRs, where smaller ligands usually bring only minor changes in the orthosteric pocket, but still result in dramatic rearrangements of the receptor helices VI and VII on the intracellular side, opening the G-protein-binding pocket (*DeVree et al., 2016*; *Rasmussen et al., 2011*; *Katritch et al., 2014*).

The extracellular motions of helices VI and VII in our models for binding of a class B agonist are likely converted into intracellular motions of these helices, corroborating the notion that a global displacement of helices VI and VII is a key feature of GPCR activation in general. While our crosslinking approach reveals details of conformational changes in the ligand-binding pocket of a class B GPCR, the recently solved cryo-EM structure of the calcitonin receptor in complex with the Gs-protein heterotrimer (PDB: 5UZ7) (*Liang et al., 2017*) provides the first structural evidence for the conformational changes occurring in the intracellular part of a class B helical bundle. Specifically, it confirms that class B GPCR activation, similar to class A (*Katritch et al., 2014*), involves a large-scale movement of helix VI, which shapes the site for G-protein binding and activation. Note, that according to our active-like CRF-CRF1R model, this helical rearrangement involves a sharp kink around the conserved G$^{6.50}$ flexible hinge in helix VI, resulting in the outward movement of both extracellular and intracellular tips of helix VI. The stability of this active-like conformation of the CRF-CRF1R complex observed in MD simulations suggests that such dramatic rearrangements can be relevant for the activation mechanism of CRF1R, and potentially of other class B GPCRs.

In summary, we implemented here a hybrid approach combining comprehensive experimental crosslinking with conformational modeling to systematically investigate the mechanism of orthosteric antagonism at a class B GPCR. We show that CRF1R peptide antagonists extensively interact with the receptor TMD, but stabilize a different conformational state in respect to the agonists. Notably, all our experimental data are obtained from the intact full-length, fully glycosylated receptor at the membrane of the live cell, and do not require a rigid conformation of the complex to determine the ligand-receptor interactions. Thus, our results provide key insights into the mechanism of GPCR activation by peptide ligands directly from a native cellular context.

## Materials and methods

### Materials

Cell lines HEK293 (Cat# ACC-305, RRID:CVCL_0045) and 293T (Cat# ACC-635, RRID:CVCL_0063) were purchased from the German Collection of Microorganisms and Cell Cultures GmbH (DSMZ). Regular testing for mycoplasma contamination is performed using the MycoAlert kit from Lonza. Dulbecco's Modified Eagle Medium (DMEM), fetal bovine serum (FBS), trypsin and penicillin–streptomycin were purchased from Gibco (Life Technologies). DNA restriction enzymes, Phusion DNA polymerase, T4 DNA ligase and PNGase F were from New England Biolabs. Plasmid preparation kits were from Macherey-Nagel. Azi was purchased from Bachem. Protease inhibitor cOmplete EDTA-free from Roche and was used supplemented with 1 mM EDTA. Polyethylenimine MAX was from Polyscience, dissolved in H$_2$O as 10 mg/mL stock solution, aliquoted and stored at −20℃. 9-Fluorenylmethoxycarbonyl (Fmoc)-protected amino acids and resin were purchased from Novabiochem or Iris Biotech. Chloroacetic acid (ClAcOH) was from Sigma.

## Peptide synthesis

Peptides were synthesized on a Syro I robot synthesizer (MultiSyntech) using a standard Fmoc/*tert*-butyl protocol (0.25 M, double coupling)(*Coin et al., 2007*). A TentaGel-S-Ram resin was used as solid support. ClAcOH was coupled to the free terminal amino group in fivefold excess together with 5 equivalents (eq.) N,N-diisopropylcarbodiimide (DIC) and 5 eq. 1-hydroxybenzotriazole (HOBt) in dimethylformamide (DMF) for 2 hr at RT. To incorporate the alkyl halide into internal positions, the original residues were exchanged to Fmoc-Lys(Dde)-OH and the last (N-terminal) amino acid was coupled as Nα-*tert*-butyloxycarbonyl(Boc)-protected. After complete assembly of the sequence on the resin, 1-(4,4-dimethyl-2,6-dioxycyclohexylidene)ethyl (Dde) was selectively removed with 3% hydrazine (v/v) in DMF (12 × 10 min) and ClAcOH was coupled in fivefold excess as described above. Peptides were cleaved from the resin with trifluoroacetic acid (TFA)/$H_2O$ (95/5 v/v, 3 hr) and precipitated in diethyl ether. Preparative RP-HPLC was performed on a C18 column (Jupiter Phenomenex 5 u, 300 Å, 250 × 10.0 mm, 5 µm) operated at 4 mL/min, using a linear gradient of acetonitrile (ACN) in $H_2O$ with 0.1% TFA from 20% to 60% in 40 min. Peptides were purified to >95%. Mass spectrometric analysis was performed on a MALDI-Tof MS instrument (Ultraflex III, Bruker). Analytic HPLC was run on each two of the following columns: I Phenomenex Jupiter C12, 4 u, 90 Å, 250 × 4.6 mm, 4 µm; II Agilent VariTide RPC, 200 Å, 250 × 4.6 mm, 6 µm; III Phenomenex Jupiter C18, 5 u, 300 Å, 250 × 4.6 mm, 5 µm; using a linear gradient of ACN in $H_2O$ with 0.1% TFA from 20% to 70% in 30 min (columns I and II) or from 20% to 70% in 40 min (columns I and III) at a flow of 0.6 mL/min (columns I and III) or 1.0 mL/min (column II).

## Cell culture and transfection

HEK293 or 293T cells were cultured in DMEM supplemented with 10% FBS and 100 U/mL penicillin-streptomycin at 37°C, 95% humidity and 5% $CO_2$. Cells were transfected at 70% confluence using 3 µg polyethylenimine (PEI) max per µg total DNA in lactate buffered saline (20 mM Na-lactate pH 4.0, 150 mM NaCl). The transfection mixture was incubated for 10 min at RT and neutralized with medium before adding it to the cells.

## cAMP accumulation assay via luciferase reporter

The day prior to transfection, $6 \times 10^6$ HEK293 cells were seeded in 10 cm dishes. Cells were co-transfected with 0.5 µg of plasmid encoding the wild-type CRF1R gene under control of a PGK promoter, 5 µg of the reporter construct (humanized PpyRE9 [*Branchini et al., 2010*] firefly luciferase gene driven by a cAMP-responsive element and followed by a PEST sequence), and 0.5 µg of plasmid encoding Renilla luciferase driven by a CMV promoter. Transfection was performed using PEI max as described above. The following day, cells were trypsinized and transferred into a 96-well plate at a density of 180,000 cells per well. 24 hr later, cells were stimulated for 3 hr at 37°C by adding serial dilutions of the ligand. Each concentration was tested in three wells. Antagonists were applied 20 min prior to agonists. After stimulation, cells were washed with ice-cold HDB (12.5 mM Hepes pH 7.4, 140 nM NaCl, 5 mM KCl) and lysed on the plate in 50 µL of luciferase buffer (10 mM $MgSO_4$, 25 mM glycylclycine, 4 mM EGTA, pH 7.8) supplemented with 1% Triton X-100 and 1 mM dithiothreitol (DTT) for 30 min on ice. The luciferase assay was performed using a BMG LABTECH Omega luminometer equipped with two injectors. 50 µL of luciferin substrate buffer (luciferase buffer supplemented with 0.3 mM luciferin, 1 mM ATP, 1 mM DTT, pH 7.8) were subsequently added to each well and the total luminescence was collected for 2 s after a 3.3 s delay. Afterwards, 50 µL of a 5 µM solution of coelenterazine in HDB were added on each well (1.67 µM final concentration of coelenterazine). The luminescence of Renilla was collected through a 475–30 nm emission filter for 2 s after a 3.8 s delay. Firefly luminescence was normalized for the Renilla luminescence. Curves were fitted by non-linear regression assuming one-site binding using Prism 5.03 for Windows (Graphpad Software Inc., San Diego, CA). $EC_{50}$ values or the percentages of receptor activation were obtained as mean ± s.e.m from at least three independent experiments, each performed in triplicate.

## Photo-crosslinking

Azi was incorporated into CRF1R using established two-plasmid transfection protocols for non-canonical amino acid mutagenesis (*Serfling and Coin, 2016*), with one plasmid encoding for the

CRF1R bearing an amber stop codon TAG at the position designated for mutation and the other plasmid encoding the orthogonal tRNA/amino-acyl tRNA synthetase (aaRS) pair dedicated to Azi. The systematic library of CRF1R-TAG mutants has been described previously (Coin et al., 2013). The plasmid encoding the translational pair contains four tandem copies of the suppressor tRNA Bst-Yam driven by the human U6 promoter and one copy of a humanized gene for the enhanced variant of the Azi-tRNA synthetase (EAziRS) driven by a PGK promoter. The humanized EAziRS gene was synthesized by Invitrogen GeneArt Gene Synthesis (Germany)/ThermoFisher Scientific (Waltham, MA).

$0.5 \times 10^6$ 293T cells per well were seeded in six-well plates. The following day, 0.5 mM Azi were added to the culture medium from a fresh 1000x stock in 0.5 M NaOH 1–2 hr prior to transfection. Cells were co-transfected with 0.5 µg of the CMV-CRF1R-FLAG TAG-mutant plasmid and 0.5 µg of the EAziRS/Bst-Yam plasmid. 48 hr after transfection, the medium was replaced by 800 µL of 100 nM peptide ligand in binding buffer (5 mM $MgCl_2$, 0.1% BSA, 0.01% Triton X-100 in HDB). The samples were incubated for 10 min at RT, followed by crosslinking on ice at 365 nm for 20 min using a BLX-365 crosslinker operated at maximal power (BioBudget Technologies, $5 \times 8$ Watt tubes). Cell were detached, transferred to 1.5-mL tubes and pelleted at $800 \times g$. Pellets were resuspended in 40 µL of HDB supplemented with 1x protease inhibitor. The cells were flash-frozen in liquid $N_2$, briefly thawed at 37°C, vortexed and centrifuged at $2500 \times g$ and 4°C for 10 min. The pellets were lysed in Triton lysis buffer (50 mM HEPES pH 7.5, 150 mM NaCl, 10% glycerol, 1% Triton X-100, 1.5 mM $MgCl_2$, 1 mM EGTA, 1 mM DTT, 1x protease inhibitor) for 30 min on ice. Insoluble debris was separated at $13,000 \times g$ and 4°C for 10 min. The supernatants were prepared for SDS-PAGE by deglycosylation using PNGase F following the supplier's instructions.

## Whole-cell ELISA

A human influenza hemagglutinin (HA) epitope was added between the endogenous cleavable signal peptide (Met1-Thr23) of CRF1R and the receptor using standard cloning methods. All mutants were cloned into pcDNA3.1.

15,000 293T cells were seeded per well of a Poly-D-lysine-coated 96-well plate. The following day, 0.5 mM Azi were added to the culture medium from a fresh 1000x stock in 0.5 M NaOH 1–2 hr prior to transfection. Cells were co-transfected with 5 ng of either HA-CRF1Rwt-FLAG or the HA-CRF1R(xxxTAG)-FLAG mutant plasmid and 5 ng of the EAziRS/Bst-Yam plasmid. The total DNA amount was filled to 100 ng per well with empty pcDNA3 vector. After 24 hr, the medium was removed and cells were fixed in 4% formaldehyde/PBS for 10 min at RT. The formaldehyde was removed by washing $3 \times 5$ min in PBS. Blocking was performed with DMEM containing 10% FBS for 1 hr at 37°C, followed by the incubation in HRP-conjugated rat-anti-HA-antibodies (Roche, clone 3F10) diluted 1:200 in DMEM containing 10% FBS for 1 hr at 37°C. Unbound antibodies were removed by washing $3 \times 5$ min in PBS. The read-out was generated by adding a freshly prepared solution of 0.7 mg/mL o-phenylenediamine dihydrochloride (OPD) and 0.08% $H_2O_2$ in 50 mM citrate-phosphate buffer. After 30 min at RT in the dark, the reaction was stopped by adding HCl to a final concentration of 170 mM and the absorption at 492 nm ($OD_{492}$) was measured. The raw data were collected as mean ± s.e.m. from three independent experiments, each performed in triplicates, and corrected for the absorption obtained from mock-transfected cells. The optimal amounts of transfected DNA and the dilution of the antibody were established through a series of preliminary experiments in which we transfected increasing amounts of CRF1Rwt plasmid (0.25–25 ng/well) and analyzed the trends of $OD_{492}$ signal obtained with variable antibody dilutions (1/100-1/5000).

## Pair-wise chemical crosslinking

$1.4 \times 10^6$ 293T cells were seeded in 6-cm dishes the day prior to transfection. Cells were transfected with 0.4 µg of the CMV-CRF1R-FLAG Cys-mutant plasmid filled up to 2 µg total DNA using an empty pcDNA3 vector. 48 hr after transfection, the cells were detached and split into up to six portions. Cells were pelleted at 800 g and resuspended in 100 µL of 100 nM peptide ligand in binding buffer (HDB, 5 mM $MgCl_2$, 0.1% BSA, 0.01% Triton X-100). The samples were incubated for 30 min at RT, detached and pelleted at $800 \times g$. Pellets were resuspended in 40 µL of HDB supplemented with 1x protease inhibitor cocktail (Roche). The cells were flash-frozen in liquid $N_2$, thawed at 37°C and centrifuged at $2500 \times g$ and 4°C for 10 min. The pellets were lysed in Triton lysis buffer (50 mM

HEPES pH 7.5, 150 mM NaCl, 10% glycerol, 1% Triton X-100, 1.5 mM $MgCl_2$, 1 mM EGTA, 1 mM DTT, 1x protease inhibitor) followed by a 30-min incubation on ice. Insoluble debris was separated at 13,000 × *g* and 4°C for 10 min. The supernatants were prepared for SDS-PAGE by incubating them for 30 min at 37°C in sample buffer (15 mM Tris-HCl pH 6.8, 0.5% SDS, 2.5% glycerol, 0.01% bromphenolblue, 150 mM DTT).

## SDS-PAGE and Western blot

Lysates were resolved on 10% polyacrylamide SDS-gels (Tris-Glycine buffered) and transferred to a PVDF membrane (Millipore Immobilon). Membranes were blocked in 5% non-fat dry milk in TBS-T (20 mM Tris-HCl, pH 7.4, 0.15 M NaCl, 0.1% Tween 20) for 1 hr at RT. The primary antibodies, either rabbit-anti-Ucn1 (PBL #5779) or rabbit-anti-CRF (PBL #rC69) were applied overnight at 4°C (1:5000 in blocking solution), followed by 3 × 10 min wash with TBS-T. The secondary antibodies, either goat-anti-rabbit IgG-HRP (SantaCruz #sc-2004) or mouse-anti-FLAG M2-HRP conjugate (Sigma #A8592) were applied for 1 hr at RT (1:15,000 or 1:5000 in blocking solution, respectively), followed by 3 × 10 min washes in TBS-T. Membranes were soaked in homemade ECL reagent (0.1 M Tris-HCl pH 8.6, 22% luminol, 10% p-coumaric acid, 10% DMSO, 0.0001% $H_2O_2$). After 1 min delay, signals were collected for 5 min in the dark (Gbox, Syngene). All Western blot results were replicated at least once with cell lysates from a second individual experiment.

## Molecular modeling

Full-length conformational models of CRF-CRF1R and $^{dFX}$CRF(12-41)-CRF1R complexes were generated with ICM-Pro molecular modeling software (www.molsoft.com) using energy-based restrained conformational modeling algorithm, similar to the procedure described in *Coin et al. (2013)*. The initial models were based on the crystal structures of the human CRF1R domains, the ECD (PDB: 3EHU)(*Pioszak et al., 2008*) and of the thermostabilized TMD (PDB: 4K5Y)(*Hollenstein et al., 2013*). Flexibility in the receptor was introduced on two levels during optimization based on the ICM Monte Carlo minimization procedure (*Abagyan and Totrov, 1994*). The whole receptor was considered flexible in the minimization runs, while extensive Monte Carlo conformational sampling was performed only for side chains located 5 Å from the peptides and specific regions of protein backbone that included the hinge region, ECL3, as well as G345$^{6.50}$ and G356$^{7.50}$ 'backbone hinges' in helices VI and VII, respectively. Flexible peptide docking and conformational sampling of the complexes was guided by experimentally derived pair-wise crosslinking restraints, which were implemented as soft harmonic potentials between the C$\beta$ of the corresponding residues with zero penalty when restraint is under 10 Å length and harmonic increase in penalty for lengths exceeding 10 Å. An exception was the dF12 residue of $^{dFX}$CRF(12-41), in which softs harmonic tethers were implemented between the N$\alpha$ of dF12 and the C$\beta$ of the corresponding residues with the penalty length of 5 Å. The restraint length was derived from the geometry of the reaction product as shown in *Figure 4*. Global energy optimizations of CRF-CRF1R and $^{dFX}$CRF(12-41)-CRF1R complexes in internal coordinates were performed by extensive conformational sampling with more than $10^7$ Monte Carlo steps. The final optimization of CRF-CRF1R and $^{dFX}$CRF(12-41)-CRF1R complexes was done with a fully flexible peptide and receptor without any distance restraints.

Additionally, we generated a full-length conformational model of the active-like CRF-CRF1R complex with incorporated conformational changes in the intracellular side of the receptor, as observed in the recently published structure of the active state complex of the calcitonin receptor with Gs heterotrimer (PDB: 5UZ7) (*Liang et al., 2017*). This model is based on the CRF-CRF1R complex, as described above, and introduces an outward shift of the intracellular part of the helix VI, mimicking the intracellular conformation of the active state calcitonin receptor structure.

## Molecular dynamics simulations

Membrane Builder module of CHARMM-GUI (*Lee et al., 2016*) server was employed to create homogenous membrane-embedded (POPC) peptide-receptor complexes within a water box, starting from the optimized peptide-receptor complex models obtained with crosslinking-restrained conformational modeling, and the system was suitably ionized to 0.15 M concentration of NaCl. Peptide-receptor complexes were pre-aligned in the OPM (Orientation of Proteins in Membranes) database (*Lomize et al., 2006*). CRF-CRF1R and $^{dFX}$CRF(12-41)-CRF1R systems of 163467 and

165848 atoms, respectively, were created, which includes 38381 TIP3 waters and 312 POPC lipids for the CRF-CRF1R system, and 38867 TIP3 waters and 320 POPC lipids for the $^{dFX}$CRF(12-41)-CRF1R system. Receptor, peptides, lipids, water molecules and ions were modeled according to CHARMM36 force field parameter set (*Best et al., 2012*). The 12 MD simulations were run, six for CRF-CRF1R and six for $^{dFX}$CRF(12-41)-CRF1R systems, under Gromacs5.0.4 (*Hess et al., 2008*) at 310 K temperature with a step size of 2 fs using six GPU-enabled nodes with 16 processors each for a period of 1 μs. Hydrogen atoms were constrained using LINCS and a cut-off of 12 Å was used for Van der Waals and short range electrostatic interactions, along with PME conditions. After minimization and equilibrations, each system was run under distance restraints between peptide and receptor, 7 restraints in case of CRF and 15 in case of $^{dFX}$CRF(12-41), for an initial period of 20 ns, and switched off afterwards. Distance restraints used harmonic penalty to the potential with 10 factor weight if distances between the pair of atoms exceeded the specified maximum value. All the distance restraints were restrained to a maximum value of 10 Å except the distance restraints between F12 of $^{dFX}$CRF(12-41) and the receptor, which were restrained to the maximum distance of 5 Å. Following the initial 20 ns of the distance restrained MD run, the distance restraints were switched off in Run2 and Run3 in both CRF-CRF1R and $^{dFX}$CRF(12-41)-CRF1R systems, to monitor evolution of the peptide-receptor complex in the absence of distance restraints. All the analyses were performed using Gromacs5.0.4 and VMD1.9.2.

## Acknowledgements

This research was supported by the Emmy-Noether Program of the Deutsche Forschungsgemeinschaft (DFG) under grant CO822/2-1 to IC, the Netherlands Organization for Scientific Research (NWO) under Rubicon grant 019.161LW.035 to BZ, and by National Institute of Health grant U54 GM094618 to VK. We thank Lei Wang (UCSF) for sharing the library of CRF1R-TAG mutants previously prepared by IC and Tingting Sun at The Salk Institute for Biological Studies, Paul Sawchenko (Peptide Biology Laboratory, The Salk Institute) for sharing the anti-CRF and anti-Ucn1 antibodies prepared by Joan Vaughan, Bruce Branchini (Connecticut College) for sharing the PpyRE9 luciferase gene. We thank Torsten Schöneberg (University of Leipzig) for his help with the ELISA, and Angela Walker for help with preparing the manuscript. The molecular dynamics simulations described in this paper were conducted on the University of Southern California's Center for High-Performance Computing clusters (hpcc.usc.edu).

## Additional information

### Funding

| Funder | Grant reference number | Author |
| --- | --- | --- |
| Deutsche Forschungsgemeinschaft | CO822/2-1 | Lisa Seidel<br>Irene Coin |
| National Institute of General Medical Sciences | U54 GM094618 | Barbara Zarzycka<br>Saheem A Zaidi<br>Vsevolod Katritch |
| Netherlands Association for Scientific Research | 019.161LW.035 | Barbara Zarzycka |

The funders had no role in study design, data collection and interpretation, or the decision to submit the work for publication.

### Author contributions

LS, Synthesized all peptides, designed and performed all photo- and pair-wise crosslinking experiments, performed the cAMP assays and ELISA, analysed the results, and wrote the manuscript with the contribution of BZ and SZ; BZ, Performed molecular modelling, analysed the results, and contributed to the manuscript; SAZ, Performed dynamics simulations, analysed the results, and contributed to the manuscript; VK, Funding acquisition, Conceived and supervised the computational part of the project, and revised the manuscript; IC, Funding acquisition, Conceived the project, supervised the biochemistry experiments, and revised the manuscript

## Author ORCIDs

Lisa Seidel, http://orcid.org/0000-0003-0020-1315
Barbara Zarzycka, http://orcid.org/0000-0002-7202-5317
Vsevolod Katritch, http://orcid.org/0000-0003-3883-4505
Irene Coin, http://orcid.org/0000-0002-7722-004X

## Additional files

### Supplementary files

• Supplementary file 1. Model with bound agonist.

• Supplementary file 2. Model with bound antagonist.

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
