## [Decision Letter]

Thank you for submitting your article "Activation of a Class B GPCR by peptide hormones: structural insight from live cells" for consideration by *eLife*. Your article has been favorably evaluated by Richard Aldrich (Senior Editor) and three reviewers, one of whom, Yibing Shan (Reviewer #1), is a member of our Board of Reviewing Editors. The following individual involved in review of your submission has agreed to reveal his identity:.

The reviewers have discussed the reviews with one another and the Reviewing Editor has drafted this decision to help you prepare a revised submission.

Summary:

Aiming to elucidate the structural changes associated with activation of Class-B GPCRs, this work used molecular dynamics simulations guided by extensive crosslinking data to construct structural models of corticotropin-releasing factor receptor type 1 (CRF1R) bound with an agonist or with an antagonist. The models suggest a conserved hinge motion and helix tilting in the vicinity of the ligand binding site.

Essential revisions:

MD simulations showed that the "compact" TMD conformation is more optimal than the "wide" conformation for CRF1R interactions with the antagonist. Movements of the extracellular parts of helices VI and VII accounts for the different conformational changes of TMD and may be a key feature of class B GPCR activation. Thus, MD simulation is critical to draw the activation mechanism of class B GPCR by peptide hormones. But the time-scale of nine MD simulations for different systems are inconsistent (Figure 7). As receptors are flexible without distance restraints, in the revision the authors should consider extend run2 and run3 to at least 1-us to provide more convincing results.

The reviewers notice that the agonist and the antagonist models are virtually identical at the cytoplasmic side. This is unlikely to be correct because conformational change at and near the ligand binding site has to be propagated to cytoplasmic side to pass the signal to downstream G proteins. Likely this is due to the simulations' failure in reaching a global equilibrium. In the revision, the models should be adaquated "relaxed" by unrestrained simulations. Other means such as slow mode analysis or coarse-grained simulation may also be appropriate to address this concern. The conformational difference in the cytoplasmic side should be clarified and discussed in the functional context.

---

## [Author Response]

*Essential revisions:*

*MD simulations showed that the "compact" TMD conformation is more optimal than the "wide" conformation for CRF1R interactions with the antagonist. Movements of the extracellular parts of helices VI and VII accounts for the different conformational changes of TMD and may be a key feature of class B GPCR activation. Thus, MD simulation is critical to draw the activation mechanism of class B GPCR by peptide hormones. But the time-scale of nine MD simulations for different systems are inconsistent (Figure 7). As receptors are flexible without distance restraints, in the revision the authors should consider extend run2 and run3 to at least 1-us to provide more convincing results.*

We have extended all the MD simulations in this work to 1 us for consistency, see Figure 7, Figure 8, Figure 9 and Figure 7—figure supplement 1.

*The reviewers notice that the agonist and the antagonist models are virtually identical at the cytoplasmic side. This is unlikely to be correct because conformational change at and near the ligand binding site has to be propagated to cytoplasmic side to pass the signal to downstream G proteins. Likely this is due to the simulations' failure in reaching a global equilibrium. In the revision, the models should be adaquated "relaxed" by unrestrained simulations. Other means such as slow mode analysis or coarse-grained simulation may also be appropriate to address this concern. The conformational difference in the cytoplasmic side should be clarified and discussed in the functional context.*

We agree with the reviewers that the intracellular side of our agonist‐bound receptor model does not reflect all the conformational changes in the active‐like receptors. While our model focuses on accurate presentation of changes in the ligand‐receptor interactions driven by experimental crosslinking data, the lack of intracellular restrains prevented accurate modeling in this part of the receptor. Moreover, the changes we observed in the intracellular side in 1us MD simulations were minor, suggesting that proper activation‐related helical rearrangement require much longer time scales as observed previously in class A receptors.

Fortunately, the very recently published electron microscopy structure of calcitonin receptor in active‐state Gs‐bound form (PDB: 5UZ7) (Liang et al., 2017) now provides a reasonable template for modeling of the intracellular rearrangements in the class B receptors. The structure shows dramatic kink and the outward bend in the helix VI around G^6.50^ flexible hinge, fully conserved in class B GPCRs. To analyze the active‐like changes in the intracellular part of CRF1R, we modified the CRF1RCRF model with a similar outward bend of helix VI in the intracellular side as observed in the calcitonin receptor structure. The modified model was subjected to unrestrained MD simulations, demonstrating its stability and persistence of the helix VI outward bend. Conservation of G^6.50^ in class B suggests that this mechanism of outward bend of both extracellular and intracellular parts of helix VI can be common for class B GPCRs. We implemented corresponding changes in Results and Discussion sections, and added Figure 8 to illustrate these new findings. We have also updated the Introduction to include the most recent structural data.